# An Odd Estimator for Shapley Values

**Fabian Fumagalli** [1] [2]  **Landon Butler** [3]  **Justin Singh Kang** [3]  **Kannan Ramchandran** [3]  **R. Teal Witter** [4]

## Abstract

The Shapley value is a ubiquitous framework for attribution in machine learning, encompassing feature importance, data valuation, and causal inference. However, its exact computation is generally intractable, necessitating efficient approximation methods. While the most effective and popular estimators leverage the *paired sampling* heuristic to reduce estimation error, the theoretical mechanism driving this improvement has remained opaque. In this work, we provide an elegant and fundamental justification for paired sampling: we prove that the Shapley value depends *exclusively* on the odd component of the set function, and that paired sampling orthogonalizes the regression objective to filter out the irrelevant even component. Leveraging this insight, we propose Odd-SHAP, a novel consistent estimator that performs polynomial regression solely on the odd subspace. By utilizing the Fourier basis to isolate this subspace and employing a proxy model to identify high-impact interactions, OddSHAP overcomes the combinatorial explosion of higher-order approximations. Through an extensive benchmark, we find that OddSHAP achieves state-of-the-art estimation accuracy at larger sampling budgets.

## 1. Introduction

As machine learning models are increasingly deployed in high-stakes domains, from healthcare and finance to criminal justice, the need to understand their decision-making processes has become paramount (Doshi-Velez & Kim, 2017). To address the opacity of these complex systems, the Shapley value has emerged as the *de facto* standard framework

for explanation. Its rigorous game-theoretic foundation has made it ubiquitous across distinct tasks, including attributing model predictions to input features (Strumbelj & Kononenko, 2010; Lundberg & Lee, 2017), quantifying the contribution of training points to model performance (Ghorbani & Zou, 2019), and estimating the causal effect of variables in structural causal models (Janzing et al., 2020; Heskes et al., 2020).

Let $d$ denote the number of players, indexed by $[d] = \{1, \ldots, d\}$. We define a value function $f : 2^{[d]} \to \mathbb{R}$ that assigns a scalar payoff $f(S)$ to every coalition $S \subseteq [d]$. This abstraction unifies various attribution tasks; depending on the context, $f(S)$ may represent: The prediction of a model masked to only include features in $S$. The loss of a model trained exclusively on the data points in $S$. The expected outcome of a structural causal model under an intervention on the variables in $S$.

In all these settings and more, the value function $f$ implicitly encodes complex higher-order interactions across the $2^d$ possible subsets. To disentangle these dynamics and attribute the total value to individual players, we turn to the Shapley value. It offers a principled attribution framework, uniquely characterized as the only method satisfying the axioms of *efficiency*, *symmetry*, *dummy*, and *linearity* (Shapley, 1953). For any player $i \in [d]$, the Shapley value of $f$ is defined as the weighted average of marginal contributions:

$$\phi_i(f) = \sum_{S \subseteq [d] \setminus \{i\}} p_{|S|}[f(S \cup \{i\}) - f(S)], \quad (1)$$

where $p_\ell = \frac{1}{d}\binom{d-1}{\ell}^{-1}$. Computing Equation (1) directly involves summing exponentially many terms, rendering exact evaluation intractable for general functions $f$.

This work specifically targets the general, model-agnostic paradigm, operating under the assumption that the target function $f$ possesses no known or accessible architecture. In this model-agnostic paradigm, the Shapley value must be estimated for medium to large values of $d$. In contrast, structural information about $f$ can be leveraged to make precise calculations mathematically tractable for all $d$ (Rozemberczki et al., 2022). When such structural blueprints are available, researchers have successfully bypassed approximation entirely by designing highly tailored, exact algorithms. Examples of these bespoke solutions span multiple archi-

[1]Department of Statistics, LMU Munich [2]MCML [3]Department of Electrical Engineering and Computer Science, UC Berkeley [4]Mathematical Sciences Department, Claremont McKenna College. Correspondence to: Fabian Fumagalli <f.fumagalli@lmu.de>, R. Teal Witter <rtealwitter@cmc.edu>.

*Proceedings of the 43rd International Conference on Machine Learning*, Seoul, South Korea. PMLR 306, 2026. Copyright 2026 by the author(s).

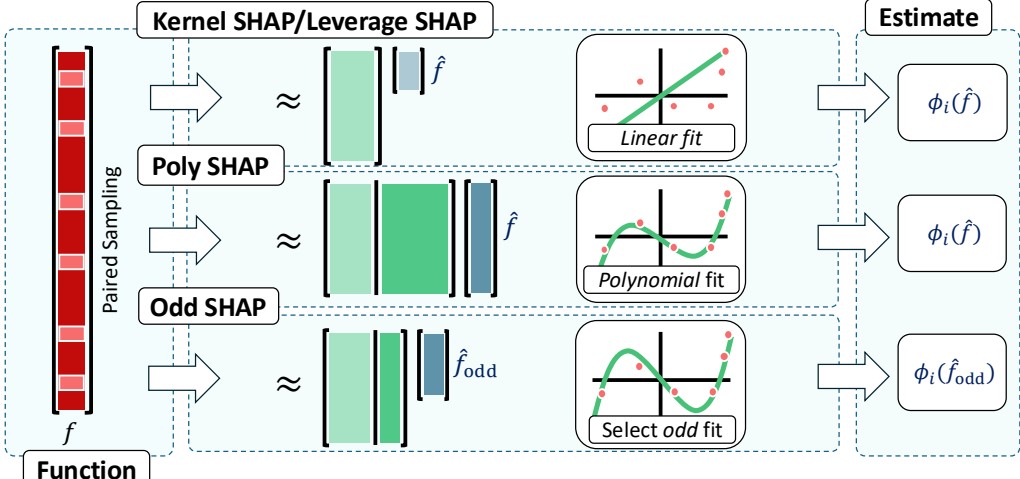

*Figure 1.* OddSHAP balances expressive power with efficiency by only fitting the *odd* component of the value function.

tectures, including K-nearest neighbor approaches for data valuation (Jia et al., 2019; Wang et al., 2023; 2024), graph neural networks (Muschalik et al., 2025), Gaussian processes (Mohammadi et al., 2025), and linear models (Strumbelj & Kononenko, 2014). The decision tree domain has similarly benefited from specialized exact methods, most notably TreeSHAP (Lundberg et al., 2020; Yu et al., 2022) and its subsequent adaptations for mapping feature interactions (Zern et al., 2023; Muschalik et al., 2024b).

### 1.1. Surrogate Estimators

The most effective estimators typically rely on function class approximations: they fit a *structured* surrogate model $\hat{f} \approx f$ and return its exact Shapley values, $\phi_i(\hat{f})$, as the estimate. Crucially, the function class must be structured enough that exactly computing the Shapley values of $\hat{f}$ is efficient.

KernelSHAP (Lundberg & Lee, 2017) and LeverageSHAP (Musco & Witter, 2025) can be viewed through this lens; they exploit a specific weighted linear regression problem whose coefficients recover the exact Shapley values (Charnes et al., 1988). Consequently, these estimators satisfy a *consistency* property: as the sample budget approaches the full coalition space ($m \to 2^d$), they converge to the true Shapley values. Although practical budgets $m$ are often orders of magnitude smaller than $2^d$, this theoretical guarantee appears to be a crucial driver of estimation performance.

Recent advancements have sought to extend this regression framework to more expressive function classes. Methods such as FourierSHAP (Gorji et al., 2025), ProxySPEX (Butler et al., 2025), and RegressionMSR (Witter et al., 2025) replace the linear basis with sparse Fourier representations or gradient boosted trees (GBTs). If the model class is correctly specified—i.e., if $f$ is truly an exact sparse Fourier function or a tree ensemble—the best approximation $\hat{f}$ co-

incides with $f$, ensuring that $\phi_i(\hat{f})$ is consistent.

However, in the general setting, $f$ rarely falls within these restricted classes. Consequently, simple proxy-based estimates are subject to model misspecification bias; even with an infinite query budget, $\phi_i(\hat{f})$ may not converge to $\phi_i(f)$. RegressionMSR addresses this limitation by introducing a residual adjustment term that makes its final estimate consistent, achieving significantly higher accuracy in the large-$m$ regime compared to unadjusted proxy methods.

Recent work has extended the linear regression strategy of KernelSHAP and LeverageSHAP to polynomial regression (Fumagalli et al., 2026). Like both linear methods, the resulting PolySHAP estimator is consistent: Fumagalli et al. (2026) show that the Shapley values of the best *polynomial* approximation to $f$, under a specific weighting and constraint, are the Shapley values of $f$ itself. While PolySHAP demonstrates superior performance by capturing higher-order interactions, it faces a scalability bottleneck; the computational cost of the regression scales quadratically with the size of the polynomial basis.

### 1.2. Our Work

Our work aims to circumvent this complexity barrier while retaining the expressive power and consistency of higher-order approximations. To achieve this, we exploit a fundamental structural property: the decomposition of the value function into odd and even components.

A set function $f$ is *odd* if $f(S) = -f(S^c)$ and *even* if $f(S) = f(S^c)$ for all $S \subseteq [d]$, where $S^c := [d] \setminus S$ is the complement. Every $f$ admits a unique decomposition into odd and even components, $f = f_{\text{odd}} + f_{\text{even}}$, with

$$f_{\text{odd}}(S) = \frac{f(S) - f(S^c)}{2}; \quad f_{\text{even}}(S) = \frac{f(S) + f(S^c)}{2}.$$

*Table 1.* Average MSE for Shapley value estimators with $m \approx 100d$. OddSHAP achieves the lowest average rank.

| | DistilBERT ($d=14$) | Estate ($d=15$) | ViT16 ($d=16$) | Cancer ($d=30$) | IL60 ($d=60$) | CG60 ($d=60$) | NHANES ($d=79$) | Crime ($d=101$) | Avg. Rank |
|---|---|---|---|---|---|---|---|---|---|
| $m$ | 1440 | 1722 | 1705 | 2281 | 5521 | 5521 | 5864 | 11126 | – |
| MSR | $7.5 \times 10^{-4}$ | $2.8 \times 10^{-2}$ | $1.2 \times 10^{-4}$ | $2.2 \times 10^{-2}$ | $3.5 \times 10^{-3}$ | $2.1 \times 10^{-3}$ | $4.7 \times 10^{-2}$ | $3.6 \times 10^{1}$ | 7.75 |
| SVARM | $3.7 \times 10^{-4}$ | $3.3 \times 10^{-2}$ | $5.7 \times 10^{-5}$ | $3.5 \times 10^{-3}$ | $2.2 \times 10^{-3}$ | $9.7 \times 10^{-4}$ | $4.0 \times 10^{-2}$ | $1.5 \times 10^{1}$ | 6.75 |
| PermutationSampling | $6.2 \times 10^{-4}$ | $5.3 \times 10^{-3}$ | $1.2 \times 10^{-4}$ | $1.1 \times 10^{-4}$ | $1.4 \times 10^{-4}$ | $1.4 \times 10^{-4}$ | $3.5 \times 10^{-3}$ | $2.7 \times 10^{0}$ | 5.62 |
| LeverageSHAP | $7.7 \times 10^{-5}$ | $3.4 \times 10^{-4}$ | $3.5 \times 10^{-5}$ | $3.2 \times 10^{-5}$ | $2.6 \times 10^{-5}$ | $2.5 \times 10^{-5}$ | $6.2 \times 10^{-4}$ | $7.5 \times 10^{-1}$ | 3.25 |
| PolySHAP-3 | $8.0 \times 10^{-5}$ | $3.2 \times 10^{-7}$ | $3.8 \times 10^{-5}$ | $4.0 \times 10^{-5}$ | | | | | 3.25 |
| RegressionMSR | $3.1 \times 10^{-5}$ | $1.3 \times 10^{-5}$ | $1.0 \times 10^{-5}$ | $3.0 \times 10^{-4}$ | $9.7 \times 10^{-6}$ | $7.6 \times 10^{-5}$ | $2.4 \times 10^{-4}$ | $5.6 \times 10^{-1}$ | 2.62 |
| Proxy | $3.6 \times 10^{-4}$ | $3.3 \times 10^{-5}$ | $4.6 \times 10^{-5}$ | $4.2 \times 10^{-4}$ | $4.9 \times 10^{-5}$ | $2.9 \times 10^{-4}$ | $7.0 \times 10^{-4}$ | $5.3 \times 10^{0}$ | 5.00 |
| FFD-RD | $1.6 \times 10^{-3}$ | $1.8 \times 10^{-6}$ | $5.3 \times 10^{-4}$ | $6.4 \times 10^{-7}$ | | | | | 5.75 |
| FFD-RD-Corrected | $1.8 \times 10^{-3}$ | | $6.8 \times 10^{-5}$ | | | | | | 8.50 |
| FourierSHAP | $1.9 \times 10^{-2}$ | $3.0 \times 10^{-2}$ | $4.9 \times 10^{-3}$ | $7.7 \times 10^{-2}$ | $4.1 \times 10^{-3}$ | $8.5 \times 10^{-3}$ | $1.7 \times 10^{-1}$ | $1.9 \times 10^{2}$ | 9.12 |
| OddSHAP | $4.6 \times 10^{-5}$ | $5.1 \times 10^{-7}$ | $1.3 \times 10^{-5}$ | $4.2 \times 10^{-6}$ | $1.6 \times 10^{-6}$ | $6.2 \times 10^{-6}$ | $5.8 \times 10^{-5}$ | $1.3 \times 10^{-1}$ | 1.50 |

Crucially, the Shapley value depends *exclusively* on the odd component of the value function:

$$\phi_i(f) = \phi_i(f_{\text{odd}}) \quad \forall i \in [d].$$

This insight suggests a targeted estimation strategy: rather than approximating the full function $f$, which entails learning potentially complex but irrelevant even structures, we can restrict our approximation to the odd component, $\hat{f}_{\text{odd}} \approx f_{\text{odd}}$, and directly compute $\phi_i(\hat{f}_{\text{odd}})$.

Remarkably, this insight clarifies the theoretical mechanism behind *paired sampling*, a popular heuristic in which every sampled coalition $S$ is paired with its complement $S^c$. While empirical studies show paired sampling greatly improves estimation performance (Covert & Lee, 2021; Mitchell et al., 2022; Olsen & Jullum, 2026), the underlying structural mechanism driving this has remained opaque. We show that paired sampling implicitly performs an even-odd decomposition, isolating the relevant signal. Mayer & Wüthrich (2025) proved that KernelSHAP with paired sampling exactly recovers Shapley values for value functions with interactions of at most degree 2, but left the analysis of higher-order interactions open. Subsequently, Fumagalli et al. (2026) demonstrated that a first-order polynomial approximation with paired sampling is equivalent to a second-order approximation, conjecturing a general pattern: that for odd $k$, an order-$k$ fit with paired sampling is equivalent to order-$(k + 1)$. We answer this conjecture in the affirmative and provide a fundamental explanation: paired sampling orthogonalizes the regression problem, separately fitting the odd and even components of $f$. Since the Shapley value of the even component vanishes, paired sampling effectively isolates the signal relevant to the Shapley value.

Beyond elucidating the mechanics of paired sampling, we leverage this decomposition to design a more precise Shapley value estimator. Our objective is to reduce the computational complexity of polynomial regression by restricting the model solely to odd terms. However, the standard *unanimity basis*—employed by KernelSHAP, LeverageSHAP, and PolySHAP—is ill-suited for this task, as its individual basis functions do not cleanly decouple into odd and even components. To overcome this, we reformulate the regression problem in the *Fourier basis*. This basis offers a decisive structural advantage: a basis function is odd if and only if its interaction order is odd. Specifically, Fourier terms of odd cardinality are odd functions, while those of even cardinality are even. Consequently, we can efficiently isolate the relevant signal by restricting the polynomial regression exclusively to odd-order Fourier terms.

Even within the restricted odd subspace, the number of candidate terms remains prohibitive; for example, there are $\binom{d}{3}$ distinct third-order interactions. Efficiently identifying the significant few is a challenge addressed by recent interaction detection methods such as SPEX (Kang et al., 2025), FourierSHAP (Gorji et al., 2025), and ProxySPEX (Butler et al., 2025), which learn sparse approximations in the Fourier domain. Leveraging the efficiency of fitting decision trees, we incorporate ProxySPEX as a selection subroutine.

Our proposed algorithm, **OddSHAP**, proceeds in two stages: first, it fits a GBT to the sampled coalitions to identify the dominant odd-order Fourier coefficients; second, it solves the polynomial regression problem restricted to this selected basis. Unlike pure proxy methods, OddSHAP produces a consistent estimator that retains the expressive power of higher-order polynomials. As demonstrated in Figure 2 and Table 1, this approach achieves state-of-the-art estimation accuracy on standard benchmarks given sufficient budgets.

Our contributions can be summarized as follows:

1. **Theoretical Unification of Paired Sampling:** Leveraging the insight that Shapley values depends strictly on the odd component of the value function, we provide the first general theoretical justification for the widely used paired sampling heuristic, proving that it implicitly decouples the regression problem into independent odd and even objectives. This result generalizes prior findings limited to low-order interactions, revealing the structural mechanism that makes this heuristic so successful in practice.

2. **The OddSHAP Estimator:** We propose OddSHAP, a consistent estimator that performs polynomial regression exclusively on a sparse selection of odd-order Fourier basis functions. By filtering out irrelevant even components and identifying high-impact interactions

via a proxy model, OddSHAP retains the expressive power of higher-order polynomials without incurring their prohibitive combinatorial cost.

3. **State-of-the-Art Performance:** Through extensive experiments on standard benchmarks, we demonstrate that OddSHAP achieves state-of-the-art estimation accuracy under larger budgets. It significantly outperforms higher-order PolySHAP in terms of computational efficiency and surpasses the prior state-of-the-art method, RegressionMSR, in estimation accuracy.

## 2. Prior Work on Regression Formulations

In this section, we review the notation and results needed to present our work. The foundational mechanism underpinning KernelSHAP, and its improved variant LeverageSHAP, is a *weighted least squares* regression. Both strategies are grounded by a rigorous theoretical guarantee: Charnes et al. (1988) proved that the coefficients of the optimal constrained weighted linear approximation $\hat{f}$ exactly recover the Shapley values of the original function $f$, i.e., $\phi_i(f) = \phi_i(\hat{f})$.

These frameworks operate using the *unanimity basis*, often referred to as the Möbius basis (Rota, 1964; Grabisch, 2016) in combinatorial contexts, where each basis function corresponds to a specific coalition $T \subseteq [d]$. Formally, the unanimity basis functions $u_T : 2^{[d]} \to \mathbb{R}$ are defined as

$$u_T(S) = \mathbb{1}[T \subseteq S].$$

Because a full basis expansion involves $2^d$ terms, fitting the complete set is computationally intractable. Therefore, we restrict our focus to a sparse subset of interactions, denoted by the collection $\mathcal{T} \subseteq 2^{[d]}$.

We define $\mathcal{F}_M(f, \mathcal{T})$ as the class of functions spanned by the unanimity basis restricted to $\mathcal{T}$, subject to boundary constraints. Specifically, every approximation $g \in \mathcal{F}_M(f, \mathcal{T})$ is constrained to match the ground truth $f$ on the empty and full coalitions, ensuring the efficiency property. Formally,

$$\mathcal{F}_M(f, \mathcal{T}) = \left\{ g = \sum_{T \in \mathcal{T}} u_T \alpha_T : g(S) = f(S), S \in \{\emptyset, [d]\} \right\}.$$

Consider the collection of empty and singleton-coalitions $\mathcal{T}_{\leq 1} = \{T \subseteq [d] : |T| \leq 1\}$. Restricting the function to this class allows exact recovery of the Shapley values via a specific weighted and constrained linear regression problem.

**Theorem 2.1** (Linear Regression (Charnes et al., 1988)).
*Let $\hat{f}$ be the best approximation in $\mathcal{F}_M(f, \mathcal{T}_{\leq 1})$ to $f$ i.e.,*

$$\hat{f} = \underset{g \in \mathcal{F}_M(f, \mathcal{T}_{\leq 1})}{\arg\min} \sum_{S \subseteq [d]: 0 < |S| < d} w_{|S|} \left(f(S) - g(S)\right)^2, \quad (2)$$

*where $w_\ell = \frac{1}{\ell(d-\ell)\binom{d}{\ell}}$. Then $\phi_i(f) = \phi_i(\hat{f})$ for all $i \in [d]$.*

See Musco & Witter (2025) for a modern proof of this result.

Because exactly solving the regression requires evaluating $f$ on exponentially many coalitions, KernelSHAP and LeverageSHAP solve an approximate regression problem on $m$ sampled coalitions. Because of Theorem 2.1, both estimators are *consistent*, i.e., they return the Shapley values as the budget $m$ approaches the total number of coalitions $2^d$.

Instead of fitting a *linear* approximation, recent Shapley value estimators use richer function classes to more accurately approximate the value function. FourierSHAP (Gorji et al., 2025) uses a sparse Fourier representation, while ProxySPEX (Butler et al., 2025) and RegressionMSR (Witter et al., 2025) use GBTs. PolySHAP (Fumagalli et al., 2026) fits a *polynomial* regression problem: like the linear regression problem, solving the weighted regression problem with higher-order basis functions also recovers the Shapley values. This has a straightforward geometric interpretation: Shapley values are a *projection* of the value function $f$ onto an affine subspace (the space of linear function satisfying the efficiency constraint), and the weighted polynomial regression is a projection onto a larger subspace. Since intermediate projections onto the larger subspace do not change the output, we have Theorem 2.2.

**Theorem 2.2** (Polynomial Regression (Fumagalli et al., 2026)). *Consider any collection of coalitions $\mathcal{T} \supseteq \mathcal{T}_{\leq 1}$. Let $\hat{f} \in \mathcal{F}_M(f, \mathcal{T})$ be the best approximation to $f$ i.e.,*

$$\hat{f} = \underset{g \in \mathcal{F}_M(f, \mathcal{T})}{\arg\min} \sum_{S \subseteq [d]: 0 < |S| < d} w_{|S|} \left(f(S) - g(S)\right)^2. \quad (3)$$

*Then $\phi_i(f) = \phi_i(\hat{f})$ for all $i \in [d]$.*

PolySHAP naturally leverages this result: Use $m$ sampled coalitions to solve an approximate version of Equation (3), then return the Shapley values of the approximation, which can be computed quickly via (Harsanyi, 1963):

$$\phi_i(\hat{f}) = \sum_{T \in \mathcal{T}: i \in T} \frac{\alpha_T}{|T|}.$$

While PolySHAP can outperform RegressionMSR for low dimensional settings with low budgets, the algorithm faces a computational bottleneck. As the number of basis vectors grows, finding the best polynomial approximation becomes more computationally intensive: The polynomial regression problem has $|\mathcal{T}|$ columns and $m$ rows, resulting in $O(|\mathcal{T}|^2 m)$ time just to solve the approximate regression problem. For example, if we select the collection up to order-$k$

$$\mathcal{T}_{\leq k} = \{T \subseteq [d] : |T| \leq k\},$$

the time complexity is $O(d^{2k} m)$. For $k \geq 2$, the resulting algorithm is prohibitively slow.

## 3. Our Odd Insights

The goal of our work is to retain the expressive power of the polynomial regression while reducing the complexity of fitting the approximation. The key is to focus on just the *odd* component of the function.

We begin with the observation that only the odd component of a value function is needed to compute its Shapley value. We can then fit an odd approximation $\hat{f}_{\text{odd}}$, retaining the expressive power of the approximation and ignoring the (potentially) expensive process of fitting the even component.

Recent work by Wendler (2023); Zhou et al. (2025); Kang et al. (2025); Gorji et al. (2025) has found that the computation of Shapley values from Fourier coefficients depends exclusively on terms of odd cardinality. As we will detail later, by properties of the Fourier basis, the following observation becomes immediate:

**Observation 3.1** (Shapley Values of Odd and Even Functions). *Let $f : 2^{[d]} \to \mathbb{R}$ be a function. Then, for all $i \in [d]$,*

$$\phi_i(f) = \phi_i(f_{\text{odd}}).$$

We provide an alternate proof of this observation, alongside all subsequent proofs, in Appendix B.

### 3.1. Paired Sampling

While interesting on its own, we can leverage Observation 3.1 to design more efficient Shapley value estimators. Instead of finding an approximation $\hat{f} \approx f$ and returning $\phi_i(\hat{f}) = \phi_i(\hat{f}_{\text{odd}})$, we will fit just the odd component $\hat{f}_{\text{odd}} \approx f_{\text{odd}}$ and directly return $\phi_i(\hat{f}_{\text{odd}})$. Intriguingly, we show that paired sampling is doing just this.

Let $\mathcal{V}$ be an unconstrained vector space of set functions, such as the span of a chosen basis. We say $\mathcal{V}$ is even-odd decomposable if any $g \in \mathcal{V}$ can be uniquely written as $g = g_{\text{odd}} + g_{\text{even}}$, where $g_{\text{odd}}, g_{\text{even}} \in \mathcal{V}$ are strictly odd and even functions, respectively.

Let $\mathcal{F}(h) \subset \mathcal{V}$ denote the affine subspace of functions constrained to match a target function $h$ on the boundaries:

$$\mathcal{F}(h) = \{g \in \mathcal{V} \mid g(\emptyset) = h(\emptyset), g([d]) = h([d])\}.$$

Linearity of the boundary evaluations ensures affine constraints decompose alongside the functions. If $g \in \mathcal{F}(f)$, its odd component satisfies $g_{\text{odd}}(\emptyset) = \frac{1}{2}(g(\emptyset) - g([d])) = \frac{1}{2}(f(\emptyset) - f([d])) = f_{\text{odd}}(\emptyset)$. Thus, $g \in \mathcal{F}(f)$ decomposes uniquely into $g_{\text{odd}} \in \mathcal{F}_{\text{odd}}(f_{\text{odd}})$ and $g_{\text{even}} \in \mathcal{F}_{\text{even}}(f_{\text{even}})$, where $\mathcal{F}_{\text{odd}}$ and $\mathcal{F}_{\text{even}}$ are the sets of odd and even functions satisfying their respective boundary constraints.

The Fourier basis discussed in the next section is clearly even-odd decomposable and, by Lemma 3.4, so is the unanimity basis when $\mathcal{T} = \mathcal{T}_{\leq k}$ for some $k$. We now consider a set $\mathcal{S}$ of $m$ paired subsets, i.e., if $S \in \mathcal{S}$ then $S^c \in \mathcal{S}$.

**Theorem 3.2** (Even-Odd Separation via Paired Sampling). *Suppose the vector space $\mathcal{V}$ is even-odd decomposable, and let $w_{|S|}$ be a symmetric weight with $w_{|S|} = w_{d-|S|}$. Under paired sampling, the weighted least squares projection onto the affine space $\mathcal{F}(f)$ completely decouples,[1]*

$$\underset{g \in \mathcal{F}(f)}{\operatorname{argmin}} \sum_{S \in \mathcal{S}} w_{|S|} \left(f(S) - g(S)\right)^2 \qquad (4)$$

$$= \underset{g_{\text{odd}} \in \mathcal{F}_{\text{odd}}(f_{\text{odd}})}{\operatorname{argmin}} \sum_{S \in \mathcal{S}} w_{|S|} \left(f_{\text{odd}}(S) - g_{\text{odd}}(S)\right)^2$$

$$+ \underset{g_{\text{even}} \in \mathcal{F}_{\text{even}}(f_{\text{even}})}{\operatorname{argmin}} \sum_{S \in \mathcal{S}} w_{|S|} \left(f_{\text{even}}(S) - g_{\text{even}}(S)\right)^2.$$

**Corollary 3.3.** *(Frontier Invariance under Unanimity) Under paired sampling, if $k$ is odd and $\hat{f}_k$ is the solution to Equation (4) under class $\mathcal{F}_M(f, \mathcal{T}_{\leq k})$, then*

$$\phi_i(\hat{f}_k) = \phi_i(\hat{f}_{k+1}) \quad \forall i \in [d].$$

This generalizes the findings of Fumagalli et al. (2026), who established this invariance when $k = 1$. The key observation is that under the unanimity basis, expanding from odd $k$ to $k + 1$ introduces terms impacting only $\mathcal{F}_{\text{even}}$.

### 3.2. Fourier Basis

Our goal is to fit only *odd* basis functions, but the unanimity basis used in KernelSHAP, LeverageSHAP, and PolySHAP consists of terms that are neither strictly odd nor even. To address this, we will fit functions in the *Fourier* basis.

The Fourier basis consists of functions $\chi_T : 2^{[d]} \to \mathbb{R}$ where

$$\chi_T(S) = (-1)^{|S \cap T|}.$$

It is easy to check that $\chi_T(S)$ is odd if $|T|$ is odd, and even if $|T|$ is even. Analogous to the unanimity basis, we define a constrained class of functions for fitting the function $f$:

$$\mathcal{F}_F(f, \mathcal{T}) = \left\{ g = \sum_{T \in \mathcal{T}} \chi_T \beta_T : g(S) = f(S), S \in \{\emptyset, [d]\} \right\}.$$

While the unanimity and Fourier bases are distinct, it has been established that their spans on coalitions up to a given order $k$ are strictly equivalent (Zhou et al., 2025, Theorem 5). Therefore, the restricted function classes coincide exactly:

**Lemma 3.4** (Unanimity and Fourier Equivalence (Grabisch, 2016; Zhou et al., 2025)). *For any $f$, the span of the unanimity and Fourier restricted classes on $\mathcal{T}_{\leq k}$ are the same:*

$$\mathcal{F}_M(f, \mathcal{T}_{\leq k}) = \mathcal{F}_F(f, \mathcal{T}_{\leq k}).$$

---

[1]When the regression solution is not unique, $+$ denotes Minkowski sum of sets, i.e. $A + B = \{g + h \mid g \in A, h \in B\}$.

---

**Algorithm 1** OddSHAP

---

**Require:** value function $f$, sampling budget $m$, regression variable factor $\eta$

$\mathcal{S} \leftarrow \text{PAIREDSAMPLE}(f, m)$

$\hat{f}_{\text{GBT}} \leftarrow$ Fit gradient boosted tree model on $\mathcal{S}$

**if** $m < d \cdot \eta$ (underdetermined regression)

  **return** $\phi_1(\hat{f}_{\text{GBT}}), \ldots, \phi_d(\hat{f}_{\text{GBT}})$ via TreeSHAP

$\mathcal{T}_{\text{odd}} \leftarrow \text{ODDINTERACTIONEXTRACT}(\hat{f}_{\text{GBT}}, \lceil \frac{m}{\eta} \rceil - d)$

$$\hat{f}_{\text{odd}} \leftarrow \operatorname*{argmin}_{g \in \mathcal{F}(f, \mathcal{T}_{\leq 1} \cup \mathcal{T}_{\text{odd}})} \sum_{S \in \mathcal{S}: 0 < |S| < d} w_{|S|} \left(f(S) - g(S)\right)^2$$

  **return** $\phi_1(\hat{f}_{\text{odd}}), \ldots, \phi_d(\hat{f}_{\text{odd}})$

---

We now show that solving the constrained, weighted regression in the Fourier basis recovers the Shapley values.

**Theorem 3.5** (Fourier Regression). *Consider any collection of coalitions $\mathcal{T} \supseteq \mathcal{T}_{\leq 1}$. Consider the best polynomial approximation to $f$ in the Fourier basis on $\mathcal{T}$:*

$$\hat{f} = \operatorname*{argmin}_{g \in \mathcal{F}_F(f, \mathcal{T})} \sum_{S \subseteq [d]: 0 < |S| < d} w_{|S|} \left(f(S) - g(S)\right)^2 \quad (5)$$

*Then $\phi_i(f) = \phi_i(\hat{f})$ for all $i \in [d]$.*

This result establishes that Fourier regression is a consistent estimator, with a computational advantage through Observation 3.1. Since the Fourier basis functions $\chi_T$ are strictly odd or even, unlike the unanimity basis functions $u_T$, we can leverage that $\phi_i(f_{\text{even}}) = 0$ to discard even basis terms. We implement this strategy in our algorithm *OddSHAP*.

## 4. OddSHAP

Algorithm 1 presents OddSHAP in three steps:

1. **Paired Sampling:** We generate a set $\mathcal{S}$ of $m$ paired subsets to facilitate even-odd separation.

2. **Interaction Screening:** We fit a GBT to the samples and extract a candidate set $\mathcal{T}_{\text{odd}}$ of odd coefficients using ProxySPEX (Butler et al., 2025), selecting those with the largest absolute magnitudes.

3. **Odd Regression:** We solve the weighted least squares problem restricted to $\mathcal{T}_{\leq 1} \cup \mathcal{T}_{\text{odd}}$ and return the exact Shapley values of the resulting approximation.

We compute the Shapley values of $\hat{f}_{\text{odd}}$ via

$$\phi_i(\hat{f}_{\text{odd}}) = -2 \sum_{\substack{T \in \mathcal{T}: i \in T \\ |T| \text{ odd}}} \frac{\beta_T}{|T|}.$$

For details of the sampling and regression, see Appendix C.

**Controlling Higher-Order Terms**  The set of odd candidate interactions, $\mathcal{T}_{\text{odd}}$, serves as the support for our subsequent regression. Recent works have demonstrated that machine-learned value functions can often be well-approximated by a small number of Fourier interactions (Kang et al., 2025; Butler et al., 2025; Gorji et al., 2025). As we later illustrate in Figure 4, this sparsity can be exploited to significantly improve Shapley value estimation. To construct $\mathcal{T}_{\text{odd}}$, following ProxySPEX (Butler et al., 2025), we fit a GBT to our samples and extract the odd-sized Fourier interactions with the highest magnitudes. To ensure a stable approximation, we require a sample-to-variable ratio of $\eta$. Because the final regression problem contains $d$ linear terms and $|\mathcal{T}_{\text{odd}}|$ higher-order terms, we set the size of the candidate set to $|\mathcal{T}_{\text{odd}}| = \lceil m/\eta \rceil - d$. This ensures the total number of regression variables strictly scales with the available sampling budget $m$. In regimes where the sampling budget $m$ is insufficient to fit even the linear terms (i.e. $m < d \cdot \eta$), we revert to the GBT estimates directly using TreeSHAP (Lundberg et al., 2020), relying on the ensemble's inherent regularization to provide a stable approximation.

**Odd Regression Details**  To efficiently compute $\hat{f}_{\text{odd}}$ in Algorithm 1, we utilize two key algorithmic optimizations that maximize numerical stability and data efficiency. First, by leveraging paired sampling, we directly pre-compute the odd targets $f_{\text{odd}}(S) = \frac{1}{2}(f(S) - f(S^c))$ for each representative of pairs in $S \in \mathcal{S}$. This filters out the even function components prior to fitting, allowing us to drop complementary pairs and solve the regression using only half of the sampled rows $(m/2)$, yielding a distinct practical speedup. Second, rather than enforcing the constraints via the numerically unstable pseudo-infinite weights common in KernelSHAP implementations (Lundberg & Lee, 2017), we solve a strictly constrained regression following Musco & Witter (2025). To obtain the solution of $\hat{f}_{\text{odd}}$, we isolate the Fourier intercept $\beta_\emptyset$ and project the design matrix onto the subspace orthogonal to the boundary constraints, guaranteeing that the estimated Shapley values sum to the exact total payoff ($f([d]) - f(\emptyset)$) regardless of the sample budget. The complete mathematical derivations of these exact Fourier boundary constraints and our projection-based optimization framework are provided in Appendix C.

**Computational Complexity**  The computational cost of OddSHAP is primarily driven by the evaluation time $T_m$ of $f$, which typically dominates when querying complex black-box functions. The subsequent estimation process involves fitting a GBT proxy and solving a sparse regression. Fitting the GBT takes $O(N_{\text{trees}} \cdot m \cdot d)$, while extracting the Fourier representation is efficient at $O(N_{\text{trees}} \cdot 4^L)$ for maximum tree depth $L$. Crucially, paired sampling allows us to directly compute the odd targets and halve the effective number of rows in our design matrix. Once the support $\mathcal{T}$ is

*Table 2.* Value functions.

| $d$ | ID | Source | Domain |
|---|---|---|---|
| 14 | DistilBERT | shapiq (Muschalik et al., 2024a) | language |
| 15 | Estate | Yeh & Hsu (2018) | tabular |
| 16 | ViT16 | shapiq (Muschalik et al., 2024a) | image |
| 30 | Cancer | Street et al. (1993) | tabular |
| 60 | CG60 | shap (Lundberg & Lee, 2017) | synthetic |
| 60 | IL60 | shap (Lundberg & Lee, 2017) | synthetic |
| 79 | NHANES | Dinh et al. (2019) | tabular |
| 101 | Crime | Redmond (2011) | tabular |

identified, solving the regression requires $O(|\mathcal{T}|^2 \frac{m}{2})$ time, followed by an $O(d \cdot |\mathcal{T}|)$ step to compute the Shapley values. By adaptively restricting the support size to follow the sample-to-variable ratio $|\mathcal{T}| = \lceil m/\eta \rceil$, OddSHAP ensures the estimation overhead scales polynomially with the budget $m$, rather than the combinatorial $O(d^k)$ scaling of the fixed-order method PolySHAP. With standard hyperparameters ($\eta = 10, L = 10$), the total runtime is approximately

$$T_m + O(N_{\text{trees}} \cdot m \cdot d + (m/\eta)^2 \cdot \frac{m}{2}),$$

keeping the method tractable even for high-dimensional inputs. See Appendix D for empirical runtime analysis.

## 5. Experiments

To empirically validate OddSHAP, we approximate Shapley values on 8 value functions for local explanations of 30 randomly selected predictions, as listed in Table 2.

**Deep Learning Value Functions**    We utilize the ViT-16 and DistilBERT value functions provided by the shapiq library (Muschalik et al., 2024a). For the image domain, the ViT-16 benchmark classifies ImageNet (Deng et al., 2009) samples using a Vision Transformer; the input is segmented into a $4 \times 4$ grid of super-patches, resulting in $d = 16$ players. For the text domain, the DistilBERT benchmark (Sanh et al., 2019) predicts sentiment on IMDB (Maas et al., 2011) review excerpts comprising $d = 14$ tokens. Both benchmarks employ baseline imputation, allowing to compute ground-truth Shapley values via exhaustive summation.

**Tabular Value Functions**    For tabular datasets, we train XGBoost classifiers (Chen & Guestrin, 2016) and define the value function via interventional feature perturbation (marginal expectation) estimated using 50 background instances. Ground-truth Shapley values are obtained using the interventional TreeSHAP algorithm (Lundberg et al., 2020).

**Flexible-Budget Estimators**    We evaluate all methods with $m$ coalitions ranging from $d + 1$ to $\min(2^d, 20000)$, which are sampled according to equal probabilities over coalition sizes and correspond to leverage scores (Musco & Witter, 2025). We compare OddSHAP-$\eta$ with $\eta = 10$

against *Permutation Sampling* (Castro et al., 2009), *SVARM* (Kolpaczki et al., 2024), *MSR* (Witter et al., 2025), which is equivalent to *Unbiased KernelSHAP* (Covert & Lee, 2021; Fumagalli et al., 2023), *FourierSHAP* (Gorji et al., 2025), and *LeverageSHAP* (Musco & Witter, 2025), which improves KernelSHAP (Lundberg & Lee, 2017) and corresponds to OddSHAP without interactions. We further compare against the proxy-based methods *RegressionMSR* (Witter et al., 2025) and Proxy (LGBM), which corresponds to RegressionMSR without adjustment, or equivalently ProxySPEX (Butler et al., 2025) using all Fourier interactions. All proxy-based methods, including ProxySPEX used by OddSHAP, use the same LightGBM (Ke et al., 2017) proxy with default configurations and maximum depth set to 10.

**Fixed-Budget Fractional Factorial Design (FFD)**    We compare our method against the fixed-budget Fractional Factorial Design (FFD) framework introduced by Zhou et al. (2025). This framework enforces a combinatorial design that restricts the evaluation overhead to a fixed $\mathcal{O}(d^2)$ total function calls. We evaluate two distinct variants: *FFD-RD* executes their baseline recursive design (RD), and *FFD-RD (corrected)* invokes their subsequent adaptive, sampling-based bias correction routine. Both variants rely on a strict truncation assumption that treats all feature interactions of order 5 and higher as exactly zero. Like OddSHAP, they exploit the algebraic property that even-order interactions completely cancel out within the Shapley value formulation.

We report *mean-squared error (MSE)* using the median across explained instances and the interquartile range (IQR). Our code is publicly available[2], and additional details and results are provided in Appendix D.

### 5.1. Approximation Quality of OddSHAP

In practice, the primary computational bottleneck for Shapley value estimation is the cost of evaluating $f$. Depending on the framework, these evaluations vary in complexity, ranging from simple model predictions (Lundberg & Lee, 2017) to the estimation of conditional distributions (Frye et al., 2021), and even complete model retraining (Strumbelj & Kononenko, 2010; Wang & Jia, 2023). Therefore, we first evaluate the performance of the approximation methods with respect to the number of queries to $f$.

Figure 2 reports the MSE of OddSHAP and the baselines with respect to the budget $m$ in log-scale. For the low-dimensional value functions (DistilBERT, Estate, and ViT16), OddSHAP yields the same performance as RegressionMSR, which is the best budget-flexible baseline. For all moderate- to high-dimensional value functions, Cancer, IL60, CG60, NHANES, and Crime, OddSHAP clearly outperforms all flexible-budget baselines when having suffi-

---

[2]https://github.com/FFmgll/oddshap

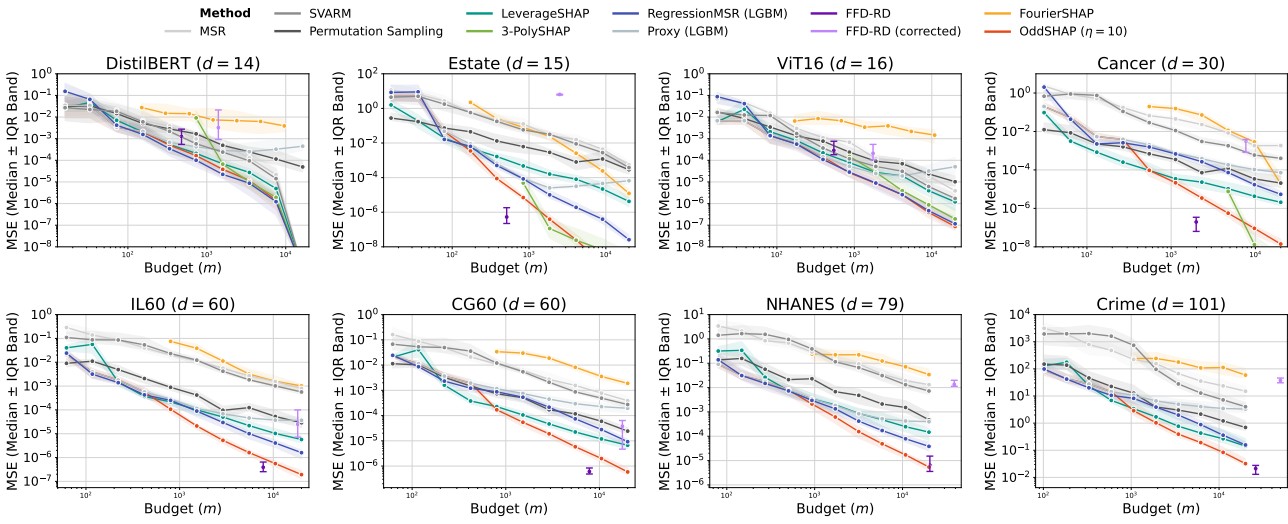

*Figure 2.* Approximation quality measured by MSE (median with interquartile range) for Shapley value estimators with paired sampling.

cient budget ($m > \eta \cdot d$) to detect and model interactions. In low sample budget scenarios, OddSHAP coincides with Proxy (LGBM), and yields comparable results with RegressionMSR or LeverageSHAP. When compared against the fixed-budget FFD framework, *FFD-RD* outperforms OddSHAP and the flexible-budget methods within equivalent budget regimes across most tree-based value functions (Estate, Cancer, IL60, and CG60). However, its performance degrades substantially on deep learning-based value functions (DistilBERT and ViT). This performance split indicates that while feature interactions of order 5 and higher are largely negligible for tree-based architectures, they remain highly active in deep neural networks, which we explicitly confirm by approximating the spectral energy in Appendix D.6. Interestingly, the adaptive *FFD-RD (corrected)* variant performs significantly worse across almost all value functions, with ViT being the sole exception. Furthermore, because the budget of FFD scales quadratically ($\mathcal{O}(d^2)$), their strict sample requirements expand drastically in high-dimensional settings, rendering them less practical as the feature dimension $d$ grows. In conclusion, OddSHAP provides a flexible-budget framework that adapts robustly across diverse interaction structures, delivering highly competitive performance across all evaluated value functions.

### 5.2. Ablation of Evaluation Costs and Runtime

To simulate diverse real-world applications, we vary the evaluation cost of $f$ as $T_1 \in \{0.001, 0.01, 0.1, 1\}$ seconds per evaluation. Although we evaluate $f$ sequentially, we note that evaluation costs can be amortized through parallelization. Figure 3 reports the MSE (median with IQR) for the runtime of all algorithms and the smallest cost ($T_1 = 0.001$s). We observe a similar pattern as in Figure 2, and provide other cost settings and datasets in Appendix D.

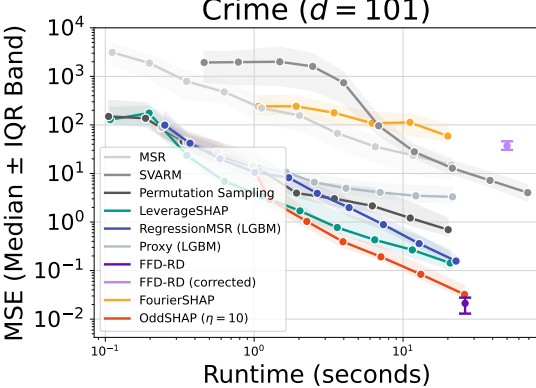

*Figure 3.* Approximation quality measured by MSE (median with interquartile range) and total runtime in seconds for a simulated setting with 0.001s per subset evaluation. Similar plots for other datasets appear in Appendix D.

### 5.3. Ablation of Interaction Sparsity

Including higher-order interactions yields a more expressive approximation of $f$, but this capacity comes at the expense of increased runtime and a larger sample budget. To evaluate this trade-off, we vary the regression variable factor across $\eta \in \{2, 5, 10, 50\}$, which controls the number of included odd interactions via $|\mathcal{T}_{\text{odd}}| = \lceil m/\eta \rceil - d$. Figure 4 illustrates the empirical impact on Shapley value accuracy. We report the MSE Ratio, defined as the Shapley MSE of OddSHAP-$\eta$ normalized by the error of the interaction-free baseline, LeverageSHAP. Estimates across all value functions were computed under a fixed budget of 10,000 samples, and we defer additional experiments under 5,000 and 20,000 samples to Appendix D, along with results for the Estate value function (omitted here due to outlier improvements).

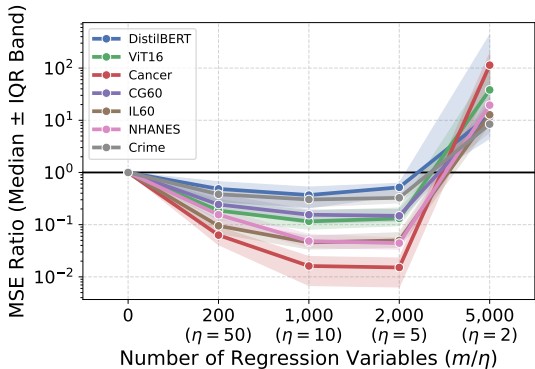

*Figure 4.* MSE ratio between OddSHAP and LeverageSHAP as a function of the number of regression variables in OddSHAP, computed under a fixed budget of 10,000 samples.

Across all datasets, we initially observe a strong decrease in the MSE Ratio as the number of interactions increases, validating that incorporating influential interactions significantly improves estimation over the interaction-free baseline. Specifically, with $\eta = 10$ (corresponding to roughly 1,000 interactions), we observe at least a $6\times$ reduction in error for all value functions and up to a $62\times$ reduction for one value function (Cancer). However, incorporating too many interactions eventually induces overfitting, reversing gains.

### 5.4. Ablation of Paired and Non-Paired Sampling

Paired sampling isolates the odd component of $f$ within the Fourier regression, eliminating the need to model even-order terms. However, it is not guaranteed to outperform non-paired sampling. Figure 5 illustrates the estimation accuracy achieved by paired and non-paired sampling across distinct interaction support. We report the MSE Ratio, defined as the Shapley MSE normalized by the error of OddSHAP (paired sampling with odd interactions). Estimates were computed under a fixed budget of 10,000 samples, and we defer additional experiments to Appendix D.

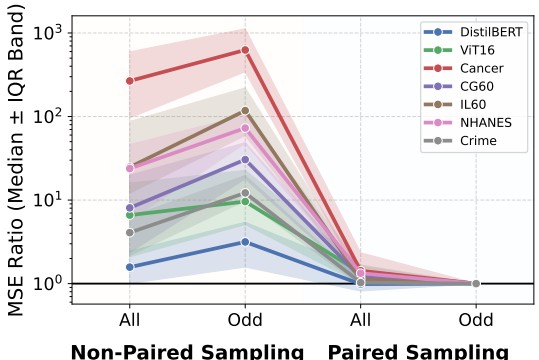

*Figure 5.* MSE ratio of paired and non-paired sampling with *All* or *Odd* interactions relative to OddSHAP under 10,000 samples.

Across all datasets, non-paired sampling underperforms paired sampling, with the non-paired *Odd* variant performing worst. Under paired sampling, including both even and odd interactions (*All*) splits the interaction budget $|\mathcal{T}| = \lceil m/\eta \rceil$ with superfluous even terms. Because these even terms mathematically cancel out, this configuration effectively restricts odd interactions to a smaller, less expressive subset, explaining its slight performance drop. Ultimately, incorporating even interactions under paired sampling only increases runtime without improving estimation.

## 6. Discussion

We introduced OddSHAP, a budget-flexible algorithm for computing Shapley values that achieves state-of-the-art performance under larger budget constraints across a broad range of feature dimensions. OddSHAP is based on a theoretical insight: Shapley values depend only on the odd component of the value function. In addition to serving as a basis for the OddSHAP algorithm, this insight provides the first rigorous theoretical basis for the popular heuristic of *paired sampling* used in prior state-of-the-art algorithms, and resolves an open conjecture in Fumagalli et al. (2026).

**Limitations and Future Directions** While OddSHAP offers clear advantages, it introduces certain computational and statistical limitations. Computationally, the algorithm scales quadratically with the number of selected interactions; since our formulation ties the interaction count to the budget $m$, the overall runtime is cubic in $m$. While the regression factor $\eta = 10$ ensures stable performance across diverse applications, we recommend capping and decoupling the number of selected interactions for larger budgets. Statistically, paired sampling does not strictly guarantee outperforming non-paired sampling. While it isolates odd interactions, reducing the regression to $m/2$ paired observations, rather than $m$ independent samples, halves the number of rows in the design matrix. This fundamentally inflates estimator variance and increases mutual coherence due to the loss of independent subset coverage. Moreover, exploring the empirical correlations between even and odd interactions observed by Butler et al. (2025) presents a promising avenue to leverage even components and improve estimation. Finally, unlike baselines such as FFD-RD that rely on rigid fixed-budget assumptions of non-existing higher-order interactions, OddSHAP accommodates more flexible interaction structures. In practice, pre-existing structural knowledge of the value function could be explicitly combined with Odd-SHAP's interaction detection to further optimize sample efficiency, a compelling direction for future exploration.

## Acknowledgments

Fabian Fumagalli gratefully acknowledges funding by the Deutsche Forschungsgemeinschaft (DFG, German Research Foundation): TRR 318/3 2026 – 438445824. This material is based upon work supported by the National Science Foundation Graduate Research Fellowship Program under Grant No. DGE-2146752. Any opinions, findings, and conclusions or recommendations expressed in this material are those of the author(s) and do not necessarily reflect the views of the National Science Foundation.

## Impact Statement

This paper presents work whose goal is to advance the field of Machine Learning. There are many potential societal consequences of our work, none of which we feel must be specifically highlighted here.

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

# A. Expanded Table

*Table 3.* Summary statistics of MSE for Shapley value estimators with $m \approx 100d$. OddSHAP achieves the lowest average rank.

| | DistilBERT ($d=14$) | Estate ($d=15$) | ViT16 ($d=16$) | Cancer ($d=30$) | IL60 ($d=60$) | CG60 ($d=60$) | NHANES ($d=79$) | Crime ($d=101$) | Avg. Rank |
|---|---|---|---|---|---|---|---|---|---|
| $m$ | 1440 | 1722 | 1705 | 2281 | 5521 | 5521 | 5864 | 11126 | – |
| **MSR** | | | | | | | | | |
| Mean | $7.5 \times 10^{-4}$ | $2.8 \times 10^{-2}$ | $1.2 \times 10^{-4}$ | $2.2 \times 10^{-2}$ | $3.5 \times 10^{-3}$ | $2.1 \times 10^{-3}$ | $4.7 \times 10^{-2}$ | $3.6 \times 10^{1}$ | 7.75 |
| 1st Quartile | $1.6 \times 10^{-4}$ | $1.7 \times 10^{-2}$ | $4.4 \times 10^{-5}$ | $1.2 \times 10^{-2}$ | $1.9 \times 10^{-3}$ | $1.1 \times 10^{-3}$ | $2.1 \times 10^{-2}$ | $9.7 \times 10^{0}$ | 7.62 |
| Median | $5.6 \times 10^{-4}$ | $2.8 \times 10^{-2}$ | $1.2 \times 10^{-4}$ | $2.3 \times 10^{-2}$ | $2.7 \times 10^{-3}$ | $1.6 \times 10^{-3}$ | $3.6 \times 10^{-2}$ | $2.4 \times 10^{1}$ | 7.88 |
| 3rd Quartile | $8.8 \times 10^{-4}$ | $4.0 \times 10^{-2}$ | $1.8 \times 10^{-4}$ | $2.9 \times 10^{-2}$ | $4.5 \times 10^{-3}$ | $2.1 \times 10^{-3}$ | $6.3 \times 10^{-2}$ | $3.7 \times 10^{1}$ | 7.75 |
| **SVARM** | | | | | | | | | |
| Mean | $3.7 \times 10^{-4}$ | $3.3 \times 10^{-2}$ | $5.7 \times 10^{-5}$ | $3.5 \times 10^{-3}$ | $2.2 \times 10^{-3}$ | $9.7 \times 10^{-4}$ | $4.0 \times 10^{-2}$ | $1.5 \times 10^{1}$ | 6.75 |
| 1st Quartile | $6.5 \times 10^{-5}$ | $1.6 \times 10^{-2}$ | $2.6 \times 10^{-5}$ | $2.3 \times 10^{-3}$ | $1.6 \times 10^{-3}$ | $7.8 \times 10^{-4}$ | $1.8 \times 10^{-2}$ | $5.3 \times 10^{0}$ | 6.50 |
| Median | $2.4 \times 10^{-4}$ | $3.2 \times 10^{-2}$ | $5.2 \times 10^{-5}$ | $3.0 \times 10^{-3}$ | $1.8 \times 10^{-3}$ | $8.7 \times 10^{-4}$ | $3.1 \times 10^{-2}$ | $7.2 \times 10^{0}$ | 6.62 |
| 3rd Quartile | $4.7 \times 10^{-4}$ | $4.8 \times 10^{-2}$ | $7.5 \times 10^{-5}$ | $4.7 \times 10^{-3}$ | $2.4 \times 10^{-3}$ | $1.1 \times 10^{-3}$ | $5.3 \times 10^{-2}$ | $1.1 \times 10^{1}$ | 6.75 |
| **PermutationSampling** | | | | | | | | | |
| Mean | $6.2 \times 10^{-4}$ | $5.3 \times 10^{-3}$ | $1.2 \times 10^{-4}$ | $1.1 \times 10^{-4}$ | $1.3 \times 10^{-4}$ | $1.4 \times 10^{-4}$ | $3.5 \times 10^{-3}$ | $2.7 \times 10^{0}$ | 5.62 |
| 1st Quartile | $2.3 \times 10^{-4}$ | $1.9 \times 10^{-3}$ | $6.2 \times 10^{-5}$ | $5.3 \times 10^{-5}$ | $9.1 \times 10^{-5}$ | $8.0 \times 10^{-5}$ | $1.3 \times 10^{-3}$ | $8.6 \times 10^{-1}$ | 5.75 |
| Median | $4.9 \times 10^{-4}$ | $2.9 \times 10^{-3}$ | $9.1 \times 10^{-5}$ | $7.2 \times 10^{-5}$ | $1.2 \times 10^{-4}$ | $1.2 \times 10^{-4}$ | $2.1 \times 10^{-3}$ | $1.2 \times 10^{0}$ | 5.62 |
| 3rd Quartile | $8.6 \times 10^{-4}$ | $4.4 \times 10^{-3}$ | $1.6 \times 10^{-4}$ | $1.7 \times 10^{-4}$ | $1.9 \times 10^{-4}$ | $1.6 \times 10^{-4}$ | $4.2 \times 10^{-3}$ | $2.6 \times 10^{0}$ | 5.62 |
| **LeverageSHAP** | | | | | | | | | |
| Mean | $7.7 \times 10^{-5}$ | $3.4 \times 10^{-4}$ | $3.5 \times 10^{-5}$ | $3.2 \times 10^{-5}$ | $2.6 \times 10^{-5}$ | $2.5 \times 10^{-5}$ | $6.2 \times 10^{-4}$ | $7.5 \times 10^{-1}$ | 3.25 |
| 1st Quartile | $3.2 \times 10^{-5}$ | $9.2 \times 10^{-5}$ | $1.9 \times 10^{-5}$ | $1.1 \times 10^{-5}$ | $1.3 \times 10^{-5}$ | $1.8 \times 10^{-5}$ | $2.4 \times 10^{-4}$ | $1.9 \times 10^{-1}$ | 3.38 |
| Median | $6.9 \times 10^{-5}$ | $1.6 \times 10^{-4}$ | $2.7 \times 10^{-5}$ | $2.3 \times 10^{-5}$ | $2.2 \times 10^{-5}$ | $2.2 \times 10^{-5}$ | $4.8 \times 10^{-4}$ | $2.7 \times 10^{-1}$ | 3.38 |
| 3rd Quartile | $1.1 \times 10^{-4}$ | $2.6 \times 10^{-4}$ | $4.3 \times 10^{-5}$ | $4.2 \times 10^{-5}$ | $3.2 \times 10^{-5}$ | $3.2 \times 10^{-5}$ | $9.0 \times 10^{-4}$ | $5.8 \times 10^{-1}$ | 3.38 |
| **PolySHAP-3** | | | | | | | | | |
| Mean | $8.0 \times 10^{-5}$ | $3.2 \times 10^{-7}$ | $3.8 \times 10^{-5}$ | $4.0 \times 10^{-5}$ | | | | | 3.25 |
| 1st Quartile | $4.5 \times 10^{-5}$ | $3.1 \times 10^{-8}$ | $1.7 \times 10^{-5}$ | $1.3 \times 10^{-5}$ | | | | | 3.25 |
| Median | $6.6 \times 10^{-5}$ | $1.1 \times 10^{-7}$ | $2.8 \times 10^{-5}$ | $2.7 \times 10^{-5}$ | | | | | 3.25 |
| 3rd Quartile | $1.1 \times 10^{-4}$ | $3.7 \times 10^{-7}$ | $4.8 \times 10^{-5}$ | $6.2 \times 10^{-5}$ | | | | | 3.25 |
| **RegressionMSR** | | | | | | | | | |
| Mean | $3.1 \times 10^{-5}$ | $1.3 \times 10^{-5}$ | $1.0 \times 10^{-5}$ | $3.0 \times 10^{-4}$ | $9.7 \times 10^{-6}$ | $7.6 \times 10^{-5}$ | $2.4 \times 10^{-4}$ | $5.6 \times 10^{-1}$ | 2.62 |
| 1st Quartile | $1.4 \times 10^{-5}$ | $6.5 \times 10^{-6}$ | $4.4 \times 10^{-6}$ | $1.8 \times 10^{-4}$ | $6.1 \times 10^{-6}$ | $5.0 \times 10^{-5}$ | $1.3 \times 10^{-4}$ | $3.0 \times 10^{-1}$ | 2.75 |
| Median | $2.2 \times 10^{-5}$ | $1.0 \times 10^{-5}$ | $8.9 \times 10^{-6}$ | $2.7 \times 10^{-4}$ | $1.0 \times 10^{-5}$ | $7.4 \times 10^{-5}$ | $1.7 \times 10^{-4}$ | $3.6 \times 10^{-1}$ | 2.88 |
| 3rd Quartile | $4.1 \times 10^{-5}$ | $1.8 \times 10^{-5}$ | $1.2 \times 10^{-5}$ | $4.1 \times 10^{-4}$ | $1.2 \times 10^{-5}$ | $1.0 \times 10^{-4}$ | $2.7 \times 10^{-4}$ | $5.4 \times 10^{-1}$ | 2.62 |
| **Proxy** | | | | | | | | | |
| Mean | $3.6 \times 10^{-4}$ | $3.3 \times 10^{-5}$ | $4.6 \times 10^{-5}$ | $4.2 \times 10^{-4}$ | $4.9 \times 10^{-5}$ | $2.9 \times 10^{-4}$ | $7.0 \times 10^{-4}$ | $5.3 \times 10^{0}$ | 5.00 |
| 1st Quartile | $1.1 \times 10^{-4}$ | $2.1 \times 10^{-5}$ | $1.4 \times 10^{-5}$ | $2.5 \times 10^{-4}$ | $3.3 \times 10^{-5}$ | $1.9 \times 10^{-4}$ | $4.1 \times 10^{-4}$ | $2.6 \times 10^{0}$ | 4.88 |
| Median | $2.9 \times 10^{-4}$ | $2.5 \times 10^{-5}$ | $2.4 \times 10^{-5}$ | $3.8 \times 10^{-4}$ | $4.6 \times 10^{-5}$ | $3.0 \times 10^{-4}$ | $5.3 \times 10^{-4}$ | $3.5 \times 10^{0}$ | 4.88 |
| 3rd Quartile | $4.8 \times 10^{-4}$ | $3.9 \times 10^{-5}$ | $4.6 \times 10^{-5}$ | $5.5 \times 10^{-4}$ | $6.1 \times 10^{-5}$ | $3.7 \times 10^{-4}$ | $9.9 \times 10^{-4}$ | $5.6 \times 10^{0}$ | 5.00 |
| **FFD-RD** | | | | | | | | | |
| Mean | $1.6 \times 10^{-3}$ | $1.8 \times 10^{-6}$ | $5.3 \times 10^{-4}$ | $6.4 \times 10^{-7}$ | | | | | 5.75 |
| 1st Quartile | $5.4 \times 10^{-4}$ | $2.2 \times 10^{-7}$ | $1.8 \times 10^{-4}$ | $6.3 \times 10^{-8}$ | | | | | 5.50 |
| Median | $1.3 \times 10^{-3}$ | $5.3 \times 10^{-7}$ | $2.9 \times 10^{-4}$ | $2.0 \times 10^{-7}$ | | | | | 5.75 |
| 3rd Quartile | $2.8 \times 10^{-3}$ | $1.8 \times 10^{-6}$ | $7.6 \times 10^{-4}$ | $3.5 \times 10^{-7}$ | | | | | 6.00 |
| **FFD-RD-Corrected** | | | | | | | | | |
| Mean | $1.8 \times 10^{-3}$ | | $6.8 \times 10^{-5}$ | | | | | | 8.50 |
| 1st Quartile | $1.8 \times 10^{-3}$ | | $6.8 \times 10^{-5}$ | | | | | | 9.50 |
| Median | $1.8 \times 10^{-3}$ | | $6.8 \times 10^{-5}$ | | | | | | 8.50 |
| 3rd Quartile | $1.8 \times 10^{-3}$ | | $6.8 \times 10^{-5}$ | | | | | | 7.50 |
| **FourierSHAP** | | | | | | | | | |
| Mean | $1.9 \times 10^{-2}$ | $3.0 \times 10^{-2}$ | $4.9 \times 10^{-3}$ | $7.7 \times 10^{-2}$ | $4.1 \times 10^{-3}$ | $8.5 \times 10^{-3}$ | $1.7 \times 10^{-1}$ | $1.9 \times 10^{2}$ | 9.12 |
| 1st Quartile | $2.9 \times 10^{-3}$ | $1.4 \times 10^{-2}$ | $2.6 \times 10^{-3}$ | $3.1 \times 10^{-2}$ | $2.0 \times 10^{-3}$ | $6.2 \times 10^{-3}$ | $8.2 \times 10^{-2}$ | $6.2 \times 10^{1}$ | 9.00 |
| Median | $7.4 \times 10^{-3}$ | $2.4 \times 10^{-2}$ | $3.4 \times 10^{-3}$ | $7.2 \times 10^{-2}$ | $3.2 \times 10^{-3}$ | $8.2 \times 10^{-3}$ | $1.2 \times 10^{-1}$ | $1.1 \times 10^{2}$ | 9.00 |
| 3rd Quartile | $2.5 \times 10^{-2}$ | $4.2 \times 10^{-2}$ | $6.4 \times 10^{-3}$ | $9.3 \times 10^{-2}$ | $5.2 \times 10^{-3}$ | $9.7 \times 10^{-3}$ | $2.3 \times 10^{-1}$ | $1.7 \times 10^{2}$ | 9.12 |
| **OddSHAP** | | | | | | | | | |
| Mean | $4.6 \times 10^{-5}$ | $5.1 \times 10^{-7}$ | $1.3 \times 10^{-5}$ | $4.2 \times 10^{-6}$ | $1.6 \times 10^{-6}$ | $6.2 \times 10^{-6}$ | $5.8 \times 10^{-5}$ | $1.3 \times 10^{-1}$ | 1.50 |
| 1st Quartile | $1.6 \times 10^{-5}$ | $2.4 \times 10^{-7}$ | $5.4 \times 10^{-6}$ | $2.5 \times 10^{-6}$ | $1.2 \times 10^{-6}$ | $4.5 \times 10^{-6}$ | $3.2 \times 10^{-5}$ | $6.6 \times 10^{-2}$ | 1.62 |
| Median | $3.8 \times 10^{-5}$ | $3.9 \times 10^{-7}$ | $8.6 \times 10^{-6}$ | $3.5 \times 10^{-6}$ | $1.6 \times 10^{-6}$ | $5.8 \times 10^{-6}$ | $4.9 \times 10^{-5}$ | $8.4 \times 10^{-2}$ | 1.38 |
| 3rd Quartile | $5.7 \times 10^{-5}$ | $7.1 \times 10^{-7}$ | $1.8 \times 10^{-5}$ | $5.4 \times 10^{-6}$ | $2.0 \times 10^{-6}$ | $7.8 \times 10^{-6}$ | $6.7 \times 10^{-5}$ | $1.3 \times 10^{-1}$ | 1.50 |

## B. Delayed Proofs

**Observation 3.1** (Shapley values of Odd and Even Functions)

*Proof.* Rearranging Equation 1, we have

$$\phi_i(f) = \sum_{S \subseteq [d]} f(S)(\mathbb{1}[i \in S]p_{|S|-1} - \mathbb{1}[i \notin S]p_{|S|}). \tag{6}$$

Since $f = f_{\text{odd}} + f_{\text{even}}$ and the Shapley value $\phi_i(\cdot)$ is a linear operator, it suffices to show that $\phi_i(f_{\text{even}}) = 0$. We will next rewrite the summation over complementary pairs $S$ and $S^c$. We pair each subset containing $i$ with its complement, which does not contain $i$. Let $S$ denote the subset in each pair where $i \in S$. Then,

$$\phi_i(f_{\text{even}}) = \sum_{S,S^c} f_{\text{even}}(S)p_{|S|-1} - f_{\text{even}}(S^c)p_{|S^c|}$$

$$= \sum_{S,S^c} f_{\text{even}}(S)[p_{|S|-1} - p_{|S^c|}], \tag{7}$$

where the last equality follows because $f_{\text{even}}$ is even (i.e., $f_{\text{even}}(S) = f_{\text{even}}(S^c)$). Notice that $|S^c| = d - |S|$. Using the symmetry of the binomial coefficient, it is easy to show that $p_{|S|-1} = p_{d-|S|}$. Then every term in Equation (7) is 0, and the lemma statement follows. $\qquad\square$

**Theorem 3.2** (Even-Odd Separation via Paired Sampling). Suppose the vector space $\mathcal{V}$ is even-odd decomposable, and let $w_{|S|}$ be a symmetric weight with $w_{|S|} = w_{d-|S|}$. Under paired sampling, the weighted least squares projection onto the affine space $\mathcal{F}(f)$ completely decouples,

$$\underset{g \in \mathcal{F}(f)}{\text{argmin}} \sum_{S \in \mathcal{S}} w_{|S|} \left(f(S) - g(S)\right)^2$$

$$= \underset{g_{\text{odd}} \in \mathcal{F}_{\text{odd}}(f_{\text{odd}})}{\text{argmin}} \sum_{S \in \mathcal{S}} w_{|S|} \left(f_{\text{odd}}(S) - g_{\text{odd}}(S)\right)^2 + \underset{g_{\text{even}} \in \mathcal{F}_{\text{even}}(f_{\text{even}})}{\text{argmin}} \sum_{S \in \mathcal{S}} w_{|S|} \left(f_{\text{even}}(S) - g_{\text{even}}(S)\right)^2.$$

*Proof of Theorem 3.2.* Consider the inner summation for a specific pair $S$ and $S^c$. Since $w_{|S|} = w_{|S^c|}$, we can expand the sum over the pair and apply an even-odd decomposition:

$$(f(S) - g(S))^2 + (f(S^c) - g(S^c))^2$$
$$= (f_{\text{even}}(S) + f_{\text{odd}}(S) - g_{\text{even}}(S) - g_{\text{odd}}(S))^2 + (f_{\text{even}}(S^c) + f_{\text{odd}}(S^c) - g_{\text{even}}(S^c) - g_{\text{odd}}(S^c))^2. \tag{8}$$

Using the definition of even and odd functions on the complement:

$$(8) = (f_{\text{even}}(S) + f_{\text{odd}}(S) - g_{\text{even}}(S) - g_{\text{odd}}(S))^2 + (f_{\text{even}}(S) - f_{\text{odd}}(S) - g_{\text{even}}(S) + g_{\text{odd}}(S))^2. \tag{9}$$

After expanding and refactoring:

$$(9) = 2\left(f_{\text{odd}}(S) - g_{\text{odd}}(S)\right)^2 + 2\left(f_{\text{even}}(S) - g_{\text{even}}(S)\right)^2$$
$$= (f_{\text{odd}}(S) - g_{\text{odd}}(S))^2 + (f_{\text{even}}(S) - g_{\text{even}}(S))^2 + (f_{\text{odd}}(S^c) - g_{\text{odd}}(S^c))^2 + (f_{\text{even}}(S^c) - g_{\text{even}}(S^c))^2,$$

where the last equation follows by again applying the definition of even and odd functions to half of the terms. Substituting this expression back into the objective and separating the minimization completes the proof. $\qquad\square$

**Corollary 3.3** (Frontier Invariance under Unanimity). Under paired sampling, if $k$ is odd and $\hat{f}_k$ is the solution to Equation (4) under class $\mathcal{F}_M(f, \mathcal{T}_{\leq k})$, then

$$\phi_i(\hat{f}_k) = \phi_i(\hat{f}_{k+1}) \quad \forall i \in [d].$$

*Proof.* By Lemma 3.4, the restricted unanimity class $\mathcal{F}_M(f, \mathcal{T}_{\leq k})$ is equivalent to the Fourier restricted class $\mathcal{F}_F(f, \mathcal{T}_{\leq k})$. We can therefore parameterize the solution space using Fourier coefficients, where the conversion to unanimity coefficients is given by Eq. 2.76 in (Grabisch, 2016):

$$\alpha_S = (-2)^{|S|} \sum_{T \supseteq S} \beta_T.$$

Consider the expansion from degree limit $k$ to $k + 1$. By the conversion above, it can be seen that this only introduces Fourier basis terms corresponding to subsets $T$ where $|T| = k + 1$. Since $k$ is odd, $k + 1$ is even. Thus, the expansion only adds dimensions to the even subspace of the Fourier domain. The subspace of odd-degree Fourier terms remains identical.

By Theorem 3.2, under paired sampling, the optimization of the odd component is independent of the even subspace. Consequently, the odd component of the minimizer remains invariant, and by Observation 3.1, $\phi(\hat{f}_k) = \phi(\hat{f}_{k+1})$. $\square$

**Lemma 3.4** (Unanimity and Fourier Equivalence). *For any $f$, the span of the unanimity and Fourier restricted classes on $\mathcal{T}_{\leq k}$ are the same i.e.,*

$$\mathcal{F}_M(f, \mathcal{T}_{\leq k}) = \mathcal{F}_F(f, \mathcal{T}_{\leq k}).$$

*Proof of Lemma 3.4.* We can use a change of variable to show this quickly. First, observe that the constraints $g(S) = f(S), S \in \{\emptyset, [d]\}$ are conditions on the function values, which are independent of the basis representation. Therefore, it suffices to show that the unconstrained spans of the two bases restricted to order $k$ are identical.

*Fourier to Unanimity:* Assume $g$ is in the unconstrained Fourier class of up to order $k$ (i.e., $\beta_T = 0$ for all $|T| > k$). To see the conversion from Fourier to unanimity coefficients, we can represent coalitions using an indicator vector $\mathbf{x} \in \{0, 1\}^d$ where $x_i = 1$ if the $i$-th feature is present. The bases can then be written as $u_T(\mathbf{x}) = \prod_{i \in T} x_i$ and $\chi_T(\mathbf{x}) = \prod_{i \in T}(1 - 2x_i)$. Expanding the Fourier basis yields $\chi_T(\mathbf{x}) = \sum_{S \subseteq T}(-2)^{|S|} u_S(\mathbf{x})$, which immediately gives the coefficient conversion formula (Grabisch, 2016):

$$\alpha_S = (-2)^{|S|} \sum_{T \supseteq S} \beta_T.$$

For any set $S$ such that $|S| > k$, the condition $T \supseteq S$ implies $|T| \geq |S| > k$. Thus, $\beta_T = 0$ for all such $T$, which implies $\alpha_S = 0$. Thus, $g$ admits a representation in the unanimity class of up to order $k$.

*Unanimity to Fourier:* Conversely, assume $g$ is in the unanimity class of up to order $k$ (i.e., $\alpha_S = 0$ for all $|S| > k$). Alternatively, we can represent coalitions using $\mathbf{x} \in \{1, -1\}^d$ where $-1$ indicates the presence of a feature. In this case, the bases are $\chi_T(\mathbf{x}) = \prod_{i \in T} x_i$ and $u_T(\mathbf{x}) = \prod_{i \in T} \frac{1 - x_i}{2}$. Expanding the unanimity basis yields $u_T(\mathbf{x}) = \frac{1}{2^{|T|}} \sum_{S \subseteq T}(-1)^{|S|} \chi_S(\mathbf{x})$, which gives the reverse conversion (Grabisch, 2016):

$$\beta_T = (-1)^{|T|} \sum_{S \supseteq T} \frac{\alpha_S}{2^{|S|}}. \tag{10}$$

Since the unconstrained subspaces are identical, and the boundary constraints are basis-independent, we conclude that $\mathcal{F}_M(f, \mathcal{T}_{\leq k}) = \mathcal{F}_F(f, \mathcal{T}_{\leq k})$. $\square$

**Theorem 3.5** (Fourier Regression). *Consider any collection of coalitions $\mathcal{T} \supset \mathcal{T}_{\leq 1}$. Consider the best polynomial approximation to $f$ in the Fourier basis on $\mathcal{T}$:*

$$\hat{f} = \operatorname*{argmin}_{g \in \mathcal{F}_F(f, \mathcal{T})} \sum_{S \subseteq [d]: 0 < |S| < d} w_{|S|} \left(f(S) - g(S)\right)^2$$

*Then $\phi_i(f) = \phi_i(\hat{f})$ for all $i \in [d]$.*

*Proof of Theorem 3.5.* The proof relies on the nested structure of the regression spaces and the property that orthogonal projections onto nested subspaces can be composed.

First, by Lemma 3.4, the restricted function classes for the unanimity and Fourier bases on $\mathcal{T}_{\leq 1}$ are identical:

$$\mathcal{F}_F(f, \mathcal{T}_{\leq 1}) = \mathcal{F}_M(f, \mathcal{T}_{\leq 1}). \tag{11}$$

Let $\mathcal{V}_{\text{small}} = \mathcal{F}_F(f, \mathcal{T}_{\leq 1})$ denote the affine subspace of additive functions satisfying the boundary constraints. Similarly, let $\mathcal{V}_{\text{large}} = \mathcal{F}_F(f, \mathcal{T})$ denote the affine subspace spanned by the Fourier basis on $\mathcal{T}$ satisfying the same constraints. Since $\mathcal{T}_{\leq 1} \subset \mathcal{T}$, it follows that $\mathcal{V}_{\text{small}} \subset \mathcal{V}_{\text{large}}$.

We define the weighted inner product $\langle g, h \rangle_w = \sum_S w_{|S|} g(S) h(S)$, where the boundary weights $w_0$ and $w_d$ can be chosen to be any arbitrary positive constants. The regression problem in Equation (5) is equivalent to finding the orthogonal projection of $f$ onto $\mathcal{V}_{\text{large}}$ under this inner product. Let $\hat{f}$ be this solution:

$$\hat{f} = \text{proj}_{\mathcal{V}_{\text{large}}}(f).$$

Similarly, let $\hat{g}$ be the best linear approximation to $f$ (the solution to Equation (2)):

$$\hat{g} = \text{proj}_{\mathcal{V}_{\text{small}}}(f).$$

By the tower property of projections onto nested subspaces, projecting $f$ onto the smaller space $\mathcal{V}_{\text{small}}$ is equivalent to first projecting $f$ onto the larger space $\mathcal{V}_{\text{large}}$, and then projecting the result onto $\mathcal{V}_{\text{small}}$. Therefore:

$$\hat{g} = \text{proj}_{\mathcal{V}_{\text{small}}}(\hat{f}). \tag{12}$$

This implies that $\hat{g}$ is *also* the best linear approximation to $\hat{f}$.

We now invoke Theorem 2.1. The theorem states that for any function, its Shapley values are exactly recovered by its best linear approximation. Applying this to $f$ and $\hat{f}$ respectively:

1. $\phi_i(f) = \phi_i(\hat{g})$ (Applying Theorem 2.1 to $f$).

2. $\phi_i(\hat{f}) = \phi_i(\hat{g})$ (Applying Theorem 2.1 to $\hat{f}$).

Combining these equalities yields $\phi_i(f) = \phi_i(\hat{f})$, completing the proof. $\qquad\square$

## C. Sampling and Regressing in OddSHAP

In this appendix, we detail the sampling strategy used in OddSHAP and derive the closed-form constraints required to enforce the efficiency property in the Fourier domain.

### C.1. Sampling Strategy

We sample coalitions $S \subseteq [d]$ without replacement, with uniform weights by coalition size. This strategy aligns naturally with the definition of the Shapley value, which equally weights coalitions uniformly by their size. Furthermore, this distribution is theoretically grounded: Musco & Witter (2025) demonstrate that these weights correspond to the leverage scores of the weighted linear regression problem in Theorem 2.1.

This sampling procedure is consistent with the standard implementations of KernelSHAP, LeverageSHAP, and PolySHAP ensuring a fair comparison between estimators.

### C.2. Exact Boundary Constraints in the Fourier Domain

A critical requirement for Shapley value estimators is the *efficiency* property, which mandates that the sum of the estimated feature attributions equals the difference between the total payout and the baseline: $\sum \phi_i = f([d]) - f(\emptyset)$. In the regression framework, this is equivalent to constraining the approximation $\hat{f}$ to match the ground truth on the empty and full sets:

$$\hat{f}(\emptyset) = f(\emptyset) \quad \text{and} \quad \hat{f}([d]) = f([d]). \tag{13}$$

Standard implementations of KernelSHAP enforce these constraints "softly" by assigning a very large weight (simulating infinity) to the samples for $\emptyset$ and $[d]$. However, this approach can lead to numerical instability and ill-conditioned design matrices. In contrast, LeverageSHAP (Musco & Witter, 2025), PolySHAP (Fumagalli et al., 2026), and our proposed OddSHAP solve the regression subject to exact constraints.

We can neatly derive these constraints for the Fourier basis. Recall that Fourier basis functions are defined as $\chi_T(S) = (-1)^{|S \cap T|}$.

**The Empty Set Constraint:** For $S = \emptyset$, the intersection $|S \cap T| = 0$ for all $T$. Thus, $\chi_T(\emptyset) = 1$ for all basis functions. The constraint $\hat{f}(\emptyset) = f(\emptyset)$ implies:

$$\sum_{T \in \mathcal{T}} \beta_T \chi_T(\emptyset) = \sum_{T \in \mathcal{T}:|T| \text{ even}} \beta_T + \sum_{T \in \mathcal{T}:|T| \text{ odd}} \beta_T = f(\emptyset). \tag{14}$$

**The Full Set Constraint:** For $S = [d]$, the intersection is simply $T$. Thus, $\chi_T([d]) = (-1)^{|T|}$. The basis function evaluates to 1 if $|T|$ is even and $-1$ if $|T|$ is odd. The constraint $\hat{f}([d]) = f([d])$ implies:

$$\sum_{T \in \mathcal{T}} \beta_T \chi_T([d]) = \sum_{T \in \mathcal{T}:|T| \text{ even}} \beta_T - \sum_{T \in \mathcal{T}:|T| \text{ odd}} \beta_T = f([d]). \tag{15}$$

In our restricted regression setting (OddSHAP), we seek to minimize the use of even-order terms. We satisfy the necessary degrees of freedom by including exactly one even coalition in our support: the empty set $T = \emptyset$. Under this construction, $\sum_{|T| \text{ even}} \beta_T$ simplifies to just $\beta_\emptyset$.

Adding Equation (14) and Equation (15) yields the solution for the intercept:

$$2\beta_\emptyset = f([d]) + f(\emptyset) \implies \beta_\emptyset = \frac{f([d]) + f(\emptyset)}{2}. \tag{16}$$

Subtracting Equation (15) from Equation (14) yields the constraint on the odd coefficients:

$$2 \sum_{T \in \mathcal{T}:|T| \text{ odd}} \beta_T = f(\emptyset) - f([d]) \implies \sum_{T \in \mathcal{T}:|T| \text{ odd}} \beta_T = -\frac{f([d]) - f(\emptyset)}{2}. \tag{17}$$

### C.3. Constrained Optimization via Projection

By explicitly fixing $\beta_\emptyset$ and deriving the sum constraint on the odd coefficients (Equation (17)), we transform the original problem into a constrained least squares problem on the odd terms only:

$$\hat{f} = \underset{\boldsymbol{\beta} \in \mathbb{R}^{|\mathcal{T}|-1} : \langle \boldsymbol{\beta}, \mathbf{1} \rangle = -\frac{f([d])-f(\emptyset)}{2}}{\operatorname{argmin}} \sum_{S \subseteq [d]: 0 < |S| < d} w_{|S|} \left( f(S) - \frac{f([d]) + f(\emptyset)}{2} - \sum_{T \in \mathcal{T} \setminus \{\emptyset\}} \chi_T(S) \beta_T \right)^2 . \tag{18}$$

We solve this efficiently by projecting the problem. Geometrically, Equation (17) restricts the solution to an affine hyperplane defined by the normal vector $\mathbf{1}$ (the all-ones vector). To handle this inhomogeneous constraint, we first shift the linear system by a particular solution that satisfies the offset. We then project the shifted regression target and design matrix onto the null space orthogonal to $\mathbf{1}$, solve the unconstrained regression in this projected subspace, and finally map the solution back by adding the particular solution to satisfy the sum constraint.

This approach guarantees that the efficiency property holds exactly, regardless of the sampling budget $m$. Even when $m \ll 2^d$, our estimator prioritizes the exact evaluation of $f(\emptyset)$ and $f([d])$, ensuring the resulting Shapley values sum to the correct total payoff.

# D. Further Experimental Results

In this section, we provide further experimental details and results. All experiments were conducted on a consumer-grade laptop with an 11th Gen Intel Core i7-11850H CPU and 30GB of RAM.

## D.1. Total Runtime of Approximation Algorithms

Figure 6 reports the total runtime (median with IQR) for all approximators across different budgets ($m$). We observe that the runtime of OddSHAP is similar to RegressionMSR, whereas MSR, LeverageSHAP, FFD-RD, FourierSHAP and Permutation Sampling are generally faster. Moreover, the runtime of OddSHAP and LeverageSHAP substantially increases for larger budgets.

Figure 7 reports the runtime in seconds for different components (sampling, proxy fit, extraction, regression) of OddSHAP across different budgets ($m$).

**Sampling**   The *sampling* is based on the `CoalitionSampler` implemented in the `shapiq` library, and uses rejection sampling. Witter et al. (2025) have proposed a more efficient method, which could optimize runtime in future implementations.

**Proxy**   Fitting the *proxy* yields a moderate increase in the runtime of OddSHAP. The LightGBM (Ke et al., 2017) proxy scales well to both, higher number of samples and higher number of features.

**Extraction**   In cases where $m < d \cdot \eta$, OddSHAP directly extract the Shapley values from the GBT, which is the main driver of computational costs. After $m$ passes this threshold, we observe that ProxySPEX efficiently extracts the most influential Fourier interactions, which results in a drop in the runtime of *extraction*. This component then stabilizes over time.

**Regression**   The *regression* component of OddSHAP is used when $m \geq d \cdot \eta$, and remains small when the number of samples is small. However, with larger budgets, the regression component becomes a substantial driver in the runtime of OddSHAP, confirming our theoretical observations. With budgets over 10,000 samples, the total runtime and regression runtime are almost identical on the log scale.

## D.2. Runtime Analysis in Simulated Settings

Figures 8 to 11 report the runtime using a simulated setting, where the cost of evaluating a single subset in $f$ was set to $T_1 \in \{0.001, 0.01, 0.1, 1\}$. Naturally, we observe that the differences observed in raw runtime from Figure 6 have stronger impact when $T_1$ is small. However, even in the lowest cost setting with $T_1 = 0.001$ we do not observe changes in the state-of-the-art performance of OddSHAP. For $T_1 = 1$, we observe curves, which are almost identical to Figure 2 with transformed x-axis.

## D.3. Ablation Study on Number of Interactions

In Figure 12, we report the ablation on varying budget $m \in \{5000, 10000, 20000\}$ samples. In all cases, particularly for the Estate value function, adding interaction terms yields performance increases compared with LeverageSHAP (OddSHAP without interactions). If $\eta$ is too small ($\eta = 2$), we observe a pattern similar to overfitting, which decreases the performance, since there is not enough budget to accurately fit each interaction term. Lastly, for larger budgets, we observe stronger performance gains using interactions in OddSHAP.

## D.4. Ablation Study on Sampling Strategies and Odd Interaction Selection

In Figure 13, we present the ablation results across varying sample budgets $m \in \{5000, 10000, 20000\}$. Across all budgets, paired sampling consistently outperforms non-paired sampling. Within the non-paired strategy, retaining only odd interactions (*Odd*) performs substantially worse than including both even and odd interactions (*All*). Conversely, under paired sampling, estimation depends entirely on the selected odd interactions; any included even interactions mathematically cancel out and merely increase computational runtime. Consequently, the paired *All* variant slightly underperforms paired *Odd*. While both configurations would yield identical estimates if given the same set of odd interactions, the *All* variant

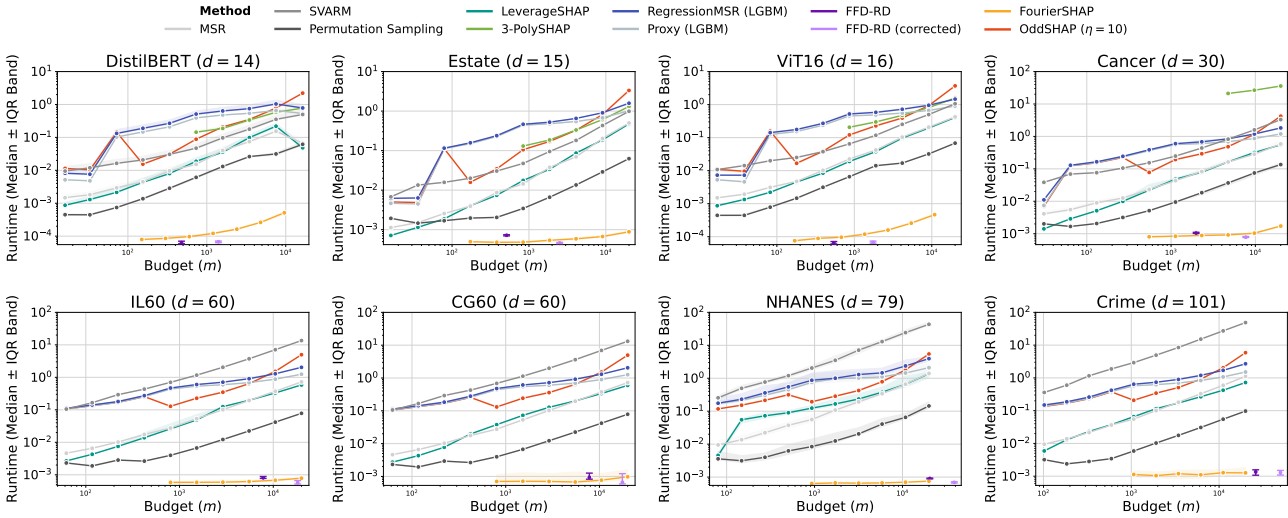

*Figure 6.* Runtime measured in seconds (median and IQR band) for varying budgets ($m$). OddSHAP has similar runtime compared with RegressionMSR, but higher runtime than LeverageSHAP.

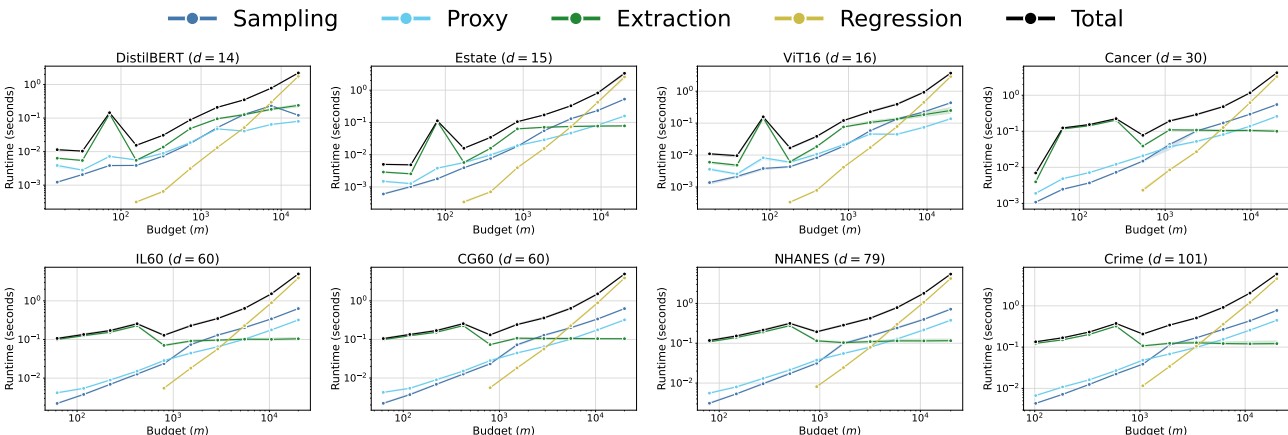

*Figure 7.* Median runtime with interquartile range (IQR) band for components of OddSHAP. All computations have a fairly small impact on runtime, where the polynomial regression is the leading factor for larger budgets.

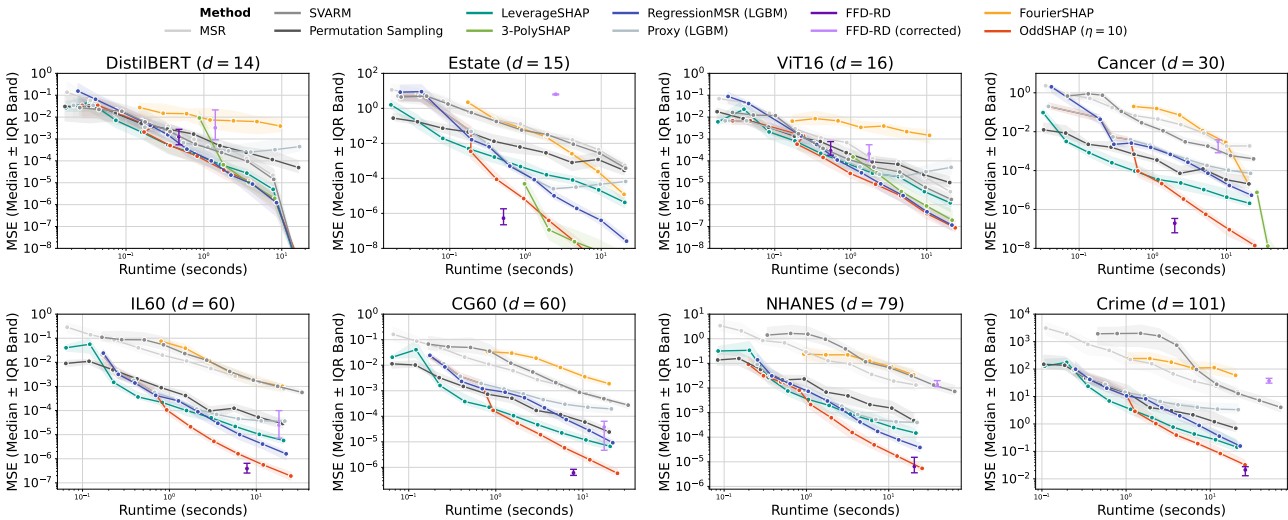

*Figure 8.* Approximation quality measured by MSE (median and interquartile range (IQR) band) for different runtimes with evaluation cost of 0.001 seconds.

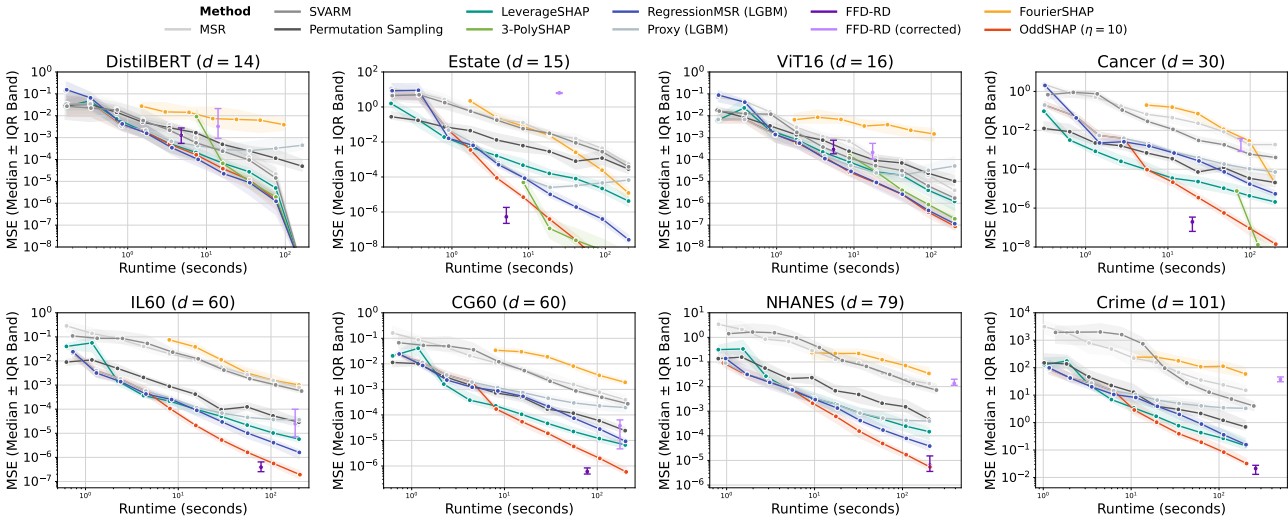

*Figure 9.* Approximation quality measured by MSE (median and interquartile range (IQR) band) for different runtimes with evaluation cost of 0.01 seconds.

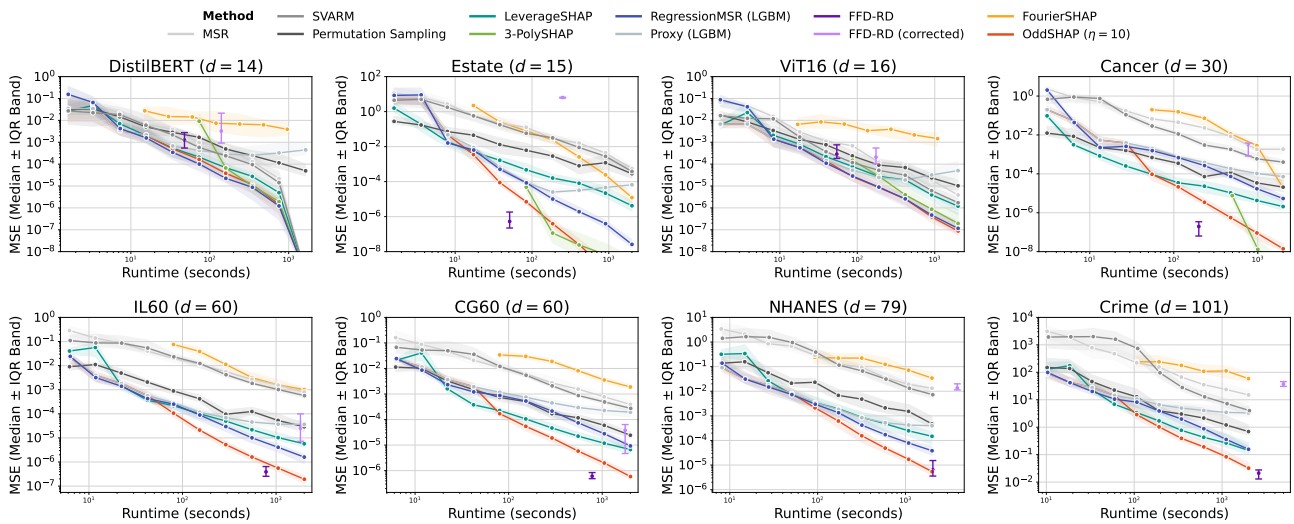

*Figure 10.* Approximation quality measured by MSE (median and interquartile range (IQR) band) for different runtimes with evaluation cost of 0.1 seconds.

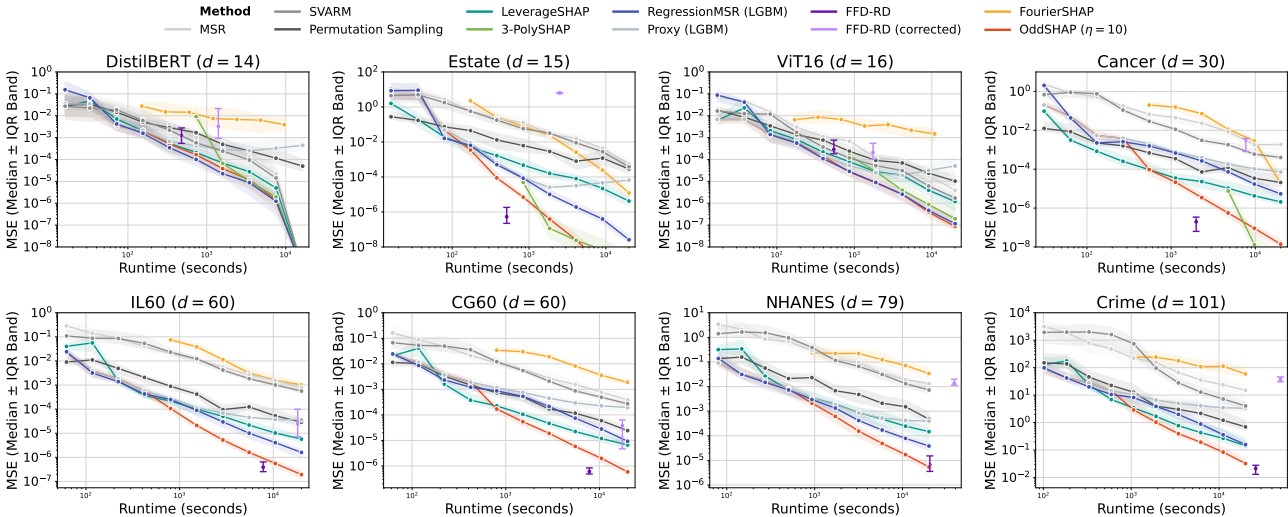

*Figure 11.* Approximation quality measured by MSE (median and interquartile range (IQR) band) for different runtimes with evaluation cost of 1 seconds.

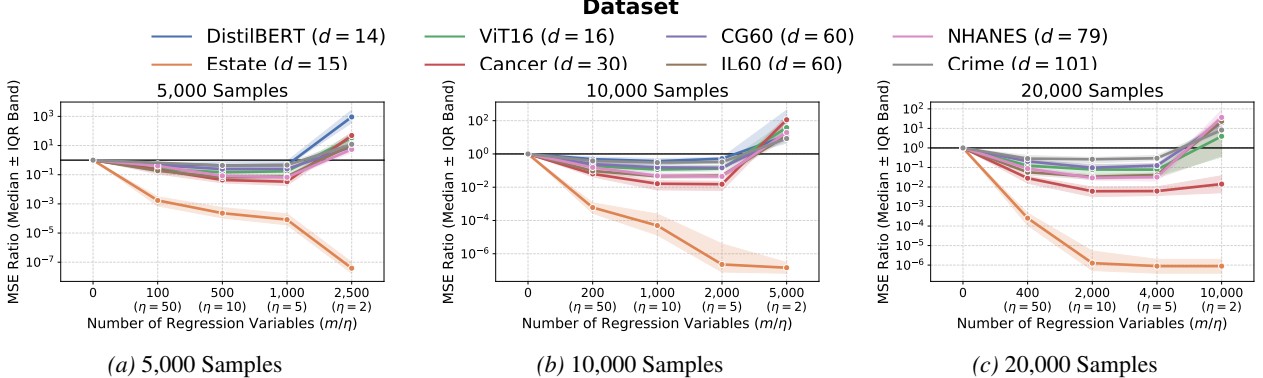

*Figure 12.* MSE Ratio (Median $\pm$ IQR Band) comparing Shapley MSE to base regression (no interactions) for different choices of $\eta$ under (left) 5,000 samples, (center) 10,000 samples, and (right) 20,000 samples.

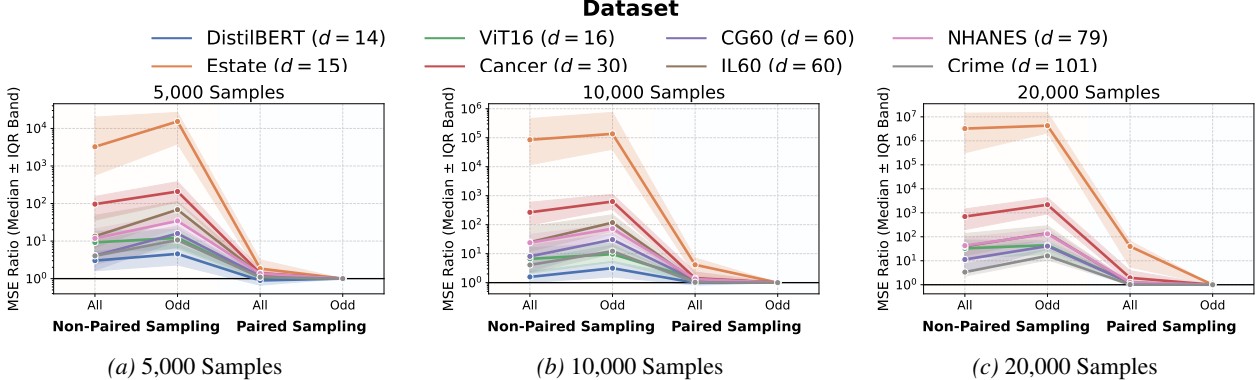

*Figure 13.* MSE Ratio (Median $\pm$ IQR Band) comparing Shapley MSE to OddSHAP for different choices of sampling (paired, non-paired) and interaction selection (all, odd only) under (left) 5,000 samples, (center) 10,000 samples, and (right) 20,000 samples.

expends part of its interaction budget on superfluous even terms, thereby reducing the number of active odd interactions and marginally degrading estimation quality.

### D.5. Ablation Study on Tree Parameters

Since OddSHAP utilizes a Gradient Boosted Tree (GBT) proxy model as a subroutine to screen for high-impact odd interactions, we conduct an ablation study to analyze how sensitive the final estimation accuracy is to the proxy's hyperparameters. Specifically, we vary the primary LightGBM configuration parameters: the maximum tree depth (`max-depth`) and the number of boosting iterations (`num-trees`). Across all main paper experiments, the default values are set to 10 and 100, respectively.

To evaluate their impact, we systematically sweep these parameters across the following configurations:

- **Maximum Depth (`max-depth`):** We vary the depth constraint within $\in \{1, 2, 5, 10, 20\}$. Altering this parameter controls the maximum interaction order that a single tree can implicitly capture during the screening phase.

- **Number of Trees (`num-trees`):** We sweep the ensemble size within $\in \{10, 50, 100, 200\}$. This adjusts the expressive capacity and regularizing capability of the proxy model.

In Figure 14 and Figure 15, we evaluate performance changes across configurations, and show the MSE ratios of the altered models relative to the default parameter baseline. Empirically, we observe that OddSHAP is remarkably robust to changes in these tree parameters, provided that the proxy is given sufficient capacity to capture higher-order dynamics (i.e., `max-depth` $\geq 5$). Restricting the depth too aggressively (e.g., `max-depth` < 5) penalizes estimation accuracy. Conversely, increasing the parameters beyond the defaults yields diminishing returns. Ultimately, these findings validate that the default configuration (`max-depth` = 10, `num-trees` = 100) is stable and generalizes effectively across all value functions. This confirms that our default configuration safely avoids overfitting while granting the proxy model sufficient capacity to capture crucial higher-order interaction signatures.

### D.6. Comparison with Fractional Factorial Designs (FFD)

To comprehensively contextualize the performance of OddSHAP, we introduce a comparison against the framework by Zhou et al. (2025), which proposes a Shapley value estimator based on Fractional Factorial Designs (FFD). Both methods share a fundamental structural philosophy: they leverage the odd-dependence property of Shapley values to eliminate the need to approximate the irrelevant even components of the value function. However, they diverge completely in their approximation mechanics. While FFD applies a non-adaptive, predefined design of experiments that fundamentally relies on a low-degree truncation assumption (positing that all interaction effects beyond a specified order are exactly zero), OddSHAP utilizes a flexible, proxy-guided interaction screening process to capture high-impact interactions dynamically, regardless of their order.

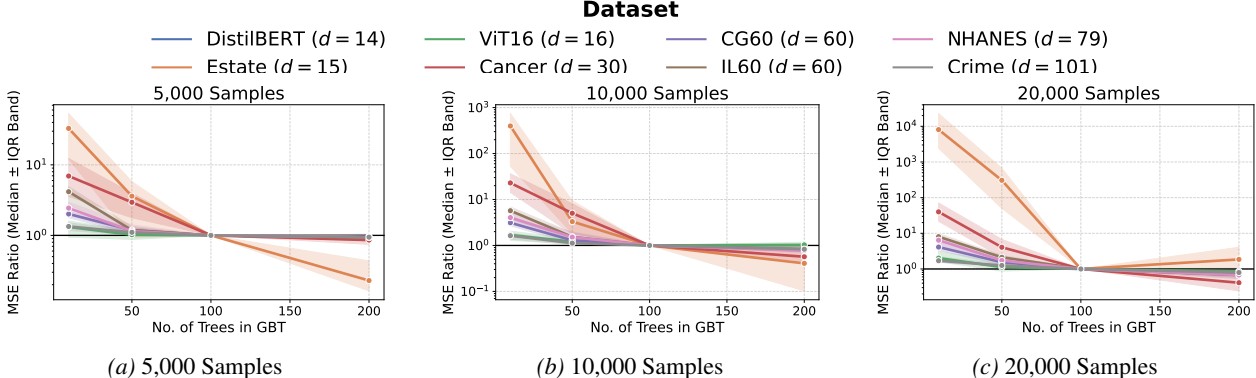

*Figure 14.* **Ablation of number of trees:** MSE Ratio (Median $\pm$ IQR Band) comparing Shapley MSE to LGBM default (`num-trees=100`) for different choices of `num-trees` under (left) 5,000 samples, (center) 10,000 samples, and (right) 20,000 samples.

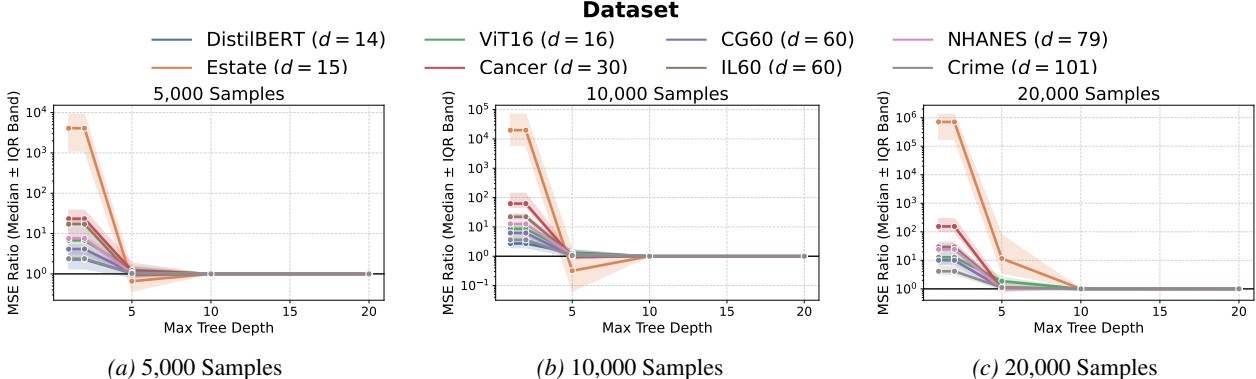

*Figure 15.* **Ablation of tree depth:** MSE Ratio (Median $\pm$ IQR Band) comparing Shapley MSE to LGBM default (`max-depth=10`) for different choices of `max-depth` under (left) 5,000 samples, (center) 10,000 samples, and (right) 20,000 samples.

**Empirical Performance Comparison**    To empirically test this assumption across our benchmark tasks, we recover the Fourier spectrum for each value function using the sparse recovery algorithm SPEX (Kang et al., 2025) under a large sample budget ($m = 100,000$). We then compute the proportion of total spectral energy (defined as the sum of squared Fourier coefficients) grouped explicitly by interaction order. The results are visualized in Figure 16.

Our benchmark evaluations uncover distinct domain-specific trade-offs between the two estimators, heavily driven by their underlying mathematical assumptions:

- **Deep Learning Architectures (ViT and DistilBERT):** On complex neural network architectures, OddSHAP consistently and significantly outperforms the FFD-RD estimator. Because models in computer vision and natural language processing naturally yield dense, long-tailed Fourier representations with substantial higher-order spectral energy, the strict truncation assumption undergirding FFD breaks down. OddSHAP remains highly robust in these regimes, as its adaptive GBT proxy screens for and isolates these crucial higher-order terms rather than ignoring them.

- **Tree-Based Models (Tabular Domain):** Conversely, on tree-based value functions (e.g., Estate, Cancer, IL60, and CG60), FFD-RD outperforms OddSHAP, while achieving comparable accuracy on the high-dimensional NHANES and Crime tasks. As demonstrated by our spectral energy analysis, our tree-based value functions yield a rapid decay in interaction terms, making the strict truncation assumption incredibly accurate for tabular datasets. In this sparse landscape, FFD's predetermined combinatorial design achieves superior data efficiency.

- **Impact of Bias Correction:** Notably, Zhou et al. (2025) propose a bias-corrected variant of their algorithm (FFD-RD with bias correction). Empirically, we find that this bias-correction step generally degrades estimation performance across almost all tasks, with the sole exception of the ViT image benchmark.

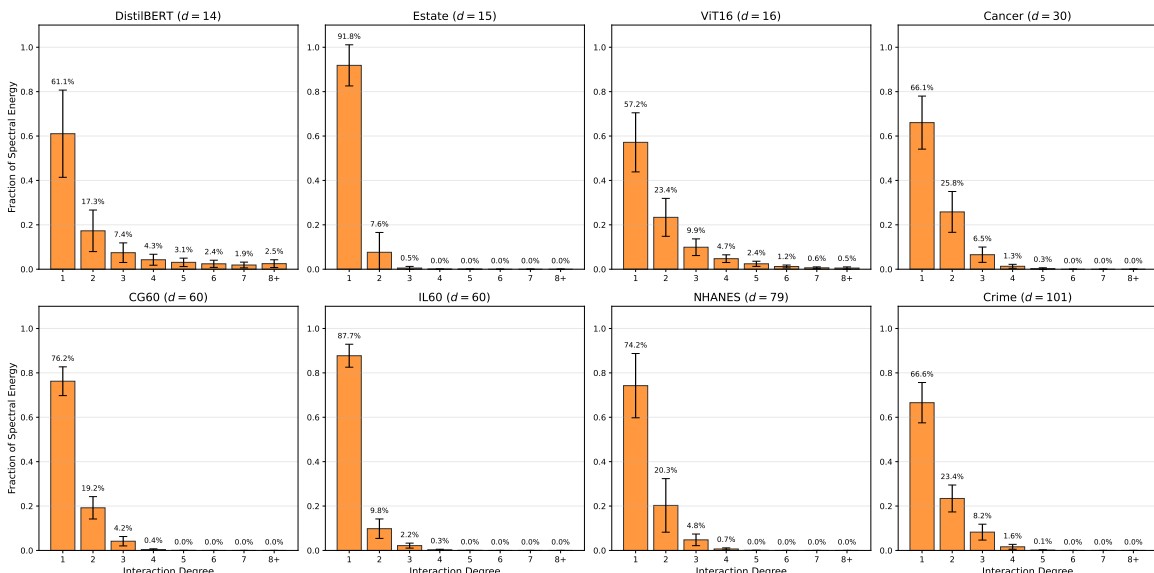

*Figure 16.* Proportion of spectral energy by interaction order. Tree-based value functions exhibit a rapid decay, with negligible energy beyond the 5th order. In contrast, deep learning architectures (DistilBERT and ViT) maintain significant higher-order spectral energy. This structural difference explains the performance degradation of FFD-RD on non-tabular domains, whereas OddSHAP remains robust across both regimes.

Ultimately, these results highlight the exact practical advantage of OddSHAP: it offers an adaptive, domain-agnostic framework that remains robust and state-of-the-art on complex deep learning architectures without suffering cataclysmic failures, while maintaining highly competitive performance in sparse tabular domains where truncation happens to hold true.

