# OpenReview forum: "An Odd Estimator for Shapley Values"
_ICML.cc/2026/Conference — ICML 2026 regular_

### Official Review · Reviewer_m9v5 · 2026-03-08

**Soundness:** 2
**Presentation:** 2
**Significance:** 2
**Originality:** 3
**Overall Recommendation:** 4
**Confidence:** 5

**Summary:**

This submission introduces an elegant even-odd decomposition to (i) shed light on the mechanism of paired sampling and (ii) design an odd estimator for approximating the Shapley value.

**Compliance With Llm Reviewing Policy:**

Affirmed.

**Final Justification:**

My concerns are fully addressed. The reason I do not go for a higher score is that the claim of resolving open questions regarding the heuristic’s effectiveness of paired sampling is an overstatement.

**Key Questions For Authors:**

- For ProxySPEX, it only specfies that the first $k$ largest coefficients are extracted. I am wondering whether the $k$ largest coefficients are selected based on their absolute values?

**Limitations:**

Generally, the sample budget $m$ is expected to be greater than $d$ to ensure a unique approximation. Then, according to Eq. (15), the time complexity is cubic in $m$, which suggests that the introduced odd estimator may not scale well for extremely large $d$.

**Strengths And Weaknesses:**

### **Strengths**
- The even-odd decomposition may be insightful for understanding the mechanism of paired sampling.
### **Weaknesses**
My major concerns are that (i) the provided proofs are neither complete or sufficient and (ii) an additional baseline may be needed.
- In the proof of Corollary 3.3, the authors simply apply the even-odd decomposition established in Theorem 3.2. The issue here is that $\mathcal{F}\_{M}(f, \mathcal{T}\_{\leq k})$ in Corollary 3.3 does not necessarily satisfy the even-odd decomposition assumption (in lines 263-267 of the left column), which is indispensable for proving Theorem 3.2. Specifically, $\mathcal{F}\_{M}(f, \mathcal{T}\_{\leq k})$ satisfies the even-odd decomposition assumption if and only if $f(\emptyset) = f([d]) = 0$, which can be certainly violated. Since Corollary 3.3 serves as one of their key contributions, this issue significantly affects the soundness of this submission.

- Let's assume Corollary 3.3 is true. Then, I think it is not strong enough to theoretically claim that paired sampling always provides better Shapley approximation, as it does not offer a direct proof showing, e.g., $\\|\hat{\phi}\^{\mathrm{paired}} - \phi\| \leq \\|\hat{\phi}\^{\mathrm{unpaired}} - \phi\\|$. With $m$ sample budget, without paired sampling, the estimate $\hat{f}$ is obtained using $m$ evaluation of $f$. By contrast, the estimate $\hat{f}\_{\mathrm{odd}}$ is obtained using $m/2$ evaluation of $f\_{\mathrm{odd}}$. It is unclear whether the gain from expanding the function class (due to paired sampling, Corollary 3.3) outweighs the negative effect of the reduction in function evaluations. Since the contributions (in lines 145-146 of the right column) claim to theoretically resolve open questions regarding the effectiveness of paired sampling, I consider this as a major concern.

- For the reason mentioned in the second point, it would be better to include a baseline that differs from the proposed odd estimators only in that (1) paired sampling is not used, and (2) the candidate set contains subsets with even coefficients extracted by ProxySPEX.

- **Minor**
  - In line 210 of the left column, where it states "In cases where the minimizer is not unique," this is confusing, because the optimal solution to (3) is known to be unique.

  - In line 208 of the left column, $w_0$ and $w_d$ are not defined, and consequently the inner product defined in line 753 is ill-defined. Nevertheless, this can be remedied by pointing out that $w_0$ and $w_d$ can be chosen to be arbitrarily positive.

  - In line 249 of the right column, unanimity functions are neither even nor odd rather than both even and odd.

  - In line 377 of the left column, MSR (Wang & Jia, 2023) is for approximating the Banzhaf value instead of the Shapley value, and it is not equal to the Unbiased KernelSHAP.

  - In line 48 of the right column, the established product-kernel functions in (Mohammadi et al., 2025a) has been found to be theoretically false in its ICLR submission.


### **Suggestions**
- It seems that the authors use $\subset$ to refer to $\subsetneq$, as they also use $\subseteq$ elsewhere. This has confused me, because people sometimes write $\subset$ to loosely mean $\subseteq$. However, Theorem 2.2 appears to suggest $\subset$ means $\subseteq$. So, I assume $\subset$ refers to $\subseteq$  in my review.

- For Eqs. (20) and (21), I did not check how the cited reference derives them, but they are quite straightforward. Using $\mathbf{x} \in \\{0, 1\\}^{d}$ to represents subsets such that $x_i = 1$ if the $i$-th feature is present, we have $u_T = \prod_{i\in T}x_i$ and $\chi_T = \prod_{i\in T} (1-2x_i)$, from which Eq. (20) follows immediately. For the other, use $\mathbf{x} \in \\{1, -1\\}^{d}$ where $-1$ indicates presence. In this case, $\chi_T = \prod_{i\in T}x_i$ and $u_T = \prod_{i\in T} \frac{1-x_i}{2}$. I think including these observations would help readers quickly verify these two conversion formulas, which also makes Lemma 3.4 immediate.

- It would be better to highlight that Theorem 3.2 holds if $w_s = w_{n-s}$ for every $s$. This is because, for KernelSHAP, $\mathcal{S}$ is sampled using $\mathbf{w}$, and all $\\{w_{|S|}\\}$ in (10) are set to $1$ when deriving the estimate.

---

> ### Author Rebuttal · Authors · 2026-03-30
>
> We thank the reviewer for their insightful and constructive feedback. We appreciate that they found our even-odd decomposition elegant. We gladly add all suggestions and fixed their minor concerns. We respond in detail below to all remaining concerns, and have run additional experiments:
> - Added baselines: https://shorturl.at/lDAYy
> - Added ablations: https://shorturl.at/lOkTy
>
> > In the proof of Corollary 3.3, the authors simply apply the even-odd decomposition established in Theorem 3.2. The issue here is that $\mathcal{F}_M$ in Corollary 3.3 does not necessarily satisfy the even-odd decomposition assumption (in lines 263--267 of the left column), which is indispensable for proving Theorem 3.2. Specifically, $\mathcal{F}_M$ satisfies the even-odd decomposition assumption if and only if $f(\emptyset) = f([d]) = 0$, which can be certainly violated.
>
> Fantastic catch, thank you! You are correct that our notation in lines 263--267 inadvertently required the odd and even components to reside in the exact same constrained space $\mathcal{F}_M$ as $f$, which mathematically forces $f(\emptyset) = f([d]) = 0$.
>
> To fix this notational issue, the decomposition must apply to the affine constraints themselves. The correction is that the constrained spaces for the decomposed components should be the even and odd functions where they match $f_{even}$ and $f_{odd}$, rather than matching $f$.
>
> With this corrected definition, the logic of Theorem 3.2 and Corollary 3.3 holds. Namely, the unconstrained span of the unanimity basis up to order $k$ is equivalent to the unconstrained span of the Fourier basis up to order $k$, which is inherently decomposable into strictly odd and even vectors. Because this underlying unconstrained space decomposes, and the linear boundary evaluations decouple perfectly alongside the function (i.e., $g_{odd}(\emptyset) = f_{odd}(\emptyset)$ and $g_{even}(\emptyset) = f_{even}(\emptyset)$), the constrained space properly satisfies the even-odd decomposition assumption. We have updated the paper to fix this.
>
> > Let's assume Corollary 3.3 is true. Then, I think it is not strong enough to theoretically claim that paired sampling always provides better Shapley approximation, as it does not offer a direct proof [...], it would be better to include a baseline that differs from the proposed odd estimators only in that (1) paired sampling is not used, and (2) the candidate set contains subsets with even coefficients extracted by ProxySPEX.
>
> Great point! While it gives an improvement for many practical Shapley value applications, paired sampling is *not* always better than unpaired sampling. We see the contribution of our work as two-fold: First, we show that paired sampling with a least squares regression estimator is equivalent to fitting only the odd component of the function. Second, with paired sampling, our OddSHAP estimator achieves the same Shapley value estimates while avoiding the computational burden of fitting the unused even component.
>
> In Figure 5 (link 2), we have run the ablation you suggested: Non-paired sampling using even and odd interactions (“All”) and odd interactions only (“Odd”). Paired sampling, using even and odd interactions (“Pairs+All”) and odd interactions only (“Pairs+Odd”=OddSHAP). All of these estimators use $m$ samples with $\lceil \frac{m}{\eta}\rceil$ interactions. Figure 5 (link 2) shows the performance of these ablated versions. We note that:
> 1. Without paired sampling, using all interactions is often better than using just the odd ones.
> 2. Empirically, paired sampling almost always helps. (But this isn’t true for every function.)
> 3. With paired sampling, using $b$ odd interactions is better than using $b$ odd and even interactions.
>
> We will add the ablations to the paper.
>
> > For ProxySPEX, it only specfies that the first largest coefficients are extracted. I am wondering whether the largest coefficients are selected based on their absolute values?
>
> Yes, we will clarify this.
>
> > Generally, the sample budget $m$ is expected to be greater than $d$ to ensure a unique approximation. Then, according to Eq. (15), the time complexity is cubic in $m$, which suggests that the introduced odd estimator may not scale well for extremely large $d$.
>
> Besides the cost of sampling from $f$, the time complexity of our estimator comes from solving the least squares problem, which depends on $m$ times the square of the number of variables. The number of variables is $d$ plus the number of higher order interactions, and we recommend using more samples than variables. Hence, as you say, the time complexity is cubic in $d$ plus the number of variables. In contrast, KernelSHAP and LeverageSHAP have a cubic dependence on $d$. Therefore, for large $d$, we suggest using a smaller number of variables, we will make this clear in the paper.

---

> > ### Author Rebuttal · Reviewer_m9v5 · 2026-04-03
> >
> > My initial impression of this submission is positive. My comments were mainly intended to have the authors further improve and complete the work. I was already aware of a straightforward fix for the issue in the proof of Corollary 3.3, which aligns with the authors’ response.  Based on these clarifications, I will raise my score accordingly.

---

### Official Review · Reviewer_u3wk · 2026-03-11

**Soundness:** 3
**Presentation:** 3
**Significance:** 3
**Originality:** 3
**Overall Recommendation:** 4
**Confidence:** 4

**Summary:**

This paper studies the problem of model-agnostic Shapley value approximation. The main idea is based on the observation that the Shapley value does not depend on the even component of the value function, but only on the odd component. Based on this observation, the authors provide a theoretical explanation that paired sampling separates the regression objective into an odd part and an even part. Also, they use the fact that, in the Fourier basis, odd and even parts can be directly separated at the basis level, and propose a new estimator called OddSHAP that focuses only on odd-order interactions. More specifically, they first select important odd-order interactions through a proxy model, and then approximate the Shapley value by performing regression only on those basis functions. In the experiments, they compare their method with existing regression-based SHAP approximation methods on several benchmarks, including text, image, and tabular data, and report strong empirical performance, especially in moderate-to-high dimensional settings.

**Compliance With Llm Reviewing Policy:**

Affirmed.

**Ethical Review Concerns:**

While reviewing this paper among the papers assigned to me, I found an overlap with another submission I was assigned, in the part discussing related works.

In Section 1 of this submission, lines 41 (first column) to 78 (second column), it is written as follows:

> Consequently, exact computation is generally feasible only when the model has exploitable structure (Rozemberczki et al., 2022). This has led to efficient, model-specific algorithms for decision trees ensembles, including TreeSHAP (Lundberg et al., 2020c; Yu et al., 2022) and interaction extensions (Zern et al., 2023; Muschalik et al., 2024c). Similar methods exist for linear (Strumbelj & Kononenko, 2014) and product-kernel models (Mohammadi et al., 2025a), Gaussian processes (Mohammadi et al., 2025b), graph neural networks (Muschalik et al., 2025), and KNN-based data valuation (Jia et al., 2019; Wang et al., 2023; 2024). Without such structure, one must rely on estimation given a query budget m to evaluate f. Because Shapley values are widely deployed, a diverse array of estimators has been developed, ranging from Monte Carlo and permutation sampling methods (Castro et al., 2009; Strumbelj & Kononenko, 2014) to regression-based approaches (Lundberg & Lee, 2017). This work focuses on this general model-agnostic setting, where no specific structure of f is assumed.

On the other hand, in another submission that I received as a reviewer, there is a paragraph in the discussion of related work that says

> Exactly computing Shapley values is generally feasible only when the underlying model possesses exploitable structure (Rozemberczki et al., 2022). This has led to efficient, model-specific algorithms for decision trees and ensembles, including TreeSHAP (Lundberg et al., 2020; Yu et al., 2022), interventional variants (Zern et al., 2023b), and extensions like TreeSHAP-IQ (Muschalik et al., 2024c). Similar exact methods exist for linear models (Strumbelj & Kononenko, 2014), product-kernel networks (Mohammadi et al., 2025a), Gaussian processes (Mohammadi et al., 2025b), graph neural networks (Muschalik et al., 2025), and KNN-based data valuation (Jia et al., 2019; Wang et al., 2023; 2024). When black-box access precludes exact methods, practitioners rely on model-agnostic estimators (Chen et al., 2023; Muschalik et al., 2024b) such as Monte Carlo sampling (Castro et al., 2009; Kolpaczki et al., 2024b;a; Fumagalli et al., 2023; Wang & Jia, 2023) or regression-based approaches like KernelSHAP (Lundberg & Lee, 2017), which has been enhanced via leverage scores (Musco & Witter, 2025) and interaction support (Fumagalli et al., 2024; Tsai et al., 2023). Finally, exact computation has recently merged with estimation via surrogate modeling (Butler et al., 2025), where auxiliary models are fit to the value function to allow efficient extraction of Shapley values (Witter et al., 2025).

As it can be seen, they overlap almost to the level of sentence structure.

**Ethical Review Flag:**

Flag this paper for an ethics review.

**Ethics Expertise Needed:**

["Research Integrity Issues (e.g., plagiarism)"]

**Final Justification:**

The authors’ rebuttal addressed the concerns that I considered important overall. I found their responses sufficiently reasonable and genuinely helpful and honest. Overall, I think this paper makes a meaningful contribution and presents convincing empirical results, so I would like to recommend acceptance.

**Key Questions For Authors:**

Below are the questions that are connected to the points I discussed in the weaknesses.

* How do you posit the novelty of the observation that even-order interactions do not matter, especially in relation to Zhou et al. (2025)?
* Compared with the FFD-based estimator of Zhou et al. (2025), what are the exact theoretical or practical advantages of OddSHAP?
* Could you add a conceptual comparison or a small empirical comparison with the method of Zhou et al. (2025), in order to show the actual difference of OddSHAP more clearly?
* The theoretical justification of paired sampling seems to be a general result that does not depend on the basis choice. Could you explain more clearly which parts require the Fourier basis and which parts do not?

Below are the questions about points that I did not discuss in the weaknesses.

* Selecting the odd-order interaction terms that are expected to be significant through GBT still seems to be heuristic to some extent. How sensitive is the method to this term selection and to hyperparameters such as tree depth or the number of trees? Could you provide a small sensitivity study?
* Could you present the full runtime more clearly, including GBT training, interaction extraction, regression solve, and SHAP computation? In particular, a runtime comparison with PolySHAP and RegressionMSR at the same $m$ and $d$ would be useful.

**Limitations:**

This paper states some technical limitations, but I think the discussion of limitations should be strengthened more explicitly, especially regarding the limitations of the GBT-based odd-subset selection step and the scope of the method when budget constraints are taken into account.

**Strengths And Weaknesses:**

## Strengths

- **Strength in presentation**

First of all, the paper itself is well written. The authors do not only present their own method, but also organize the development of the regression-based SHAP approximation literature in a way that is not too hard to follow. Because of this, I think the paper can also serve as a useful entry point for readers who are new to this line of work.

- **Two-fold theoretical strengths**

The most notable theoretical strength is that the paper gives a theoretical justification for paired sampling through this odd-even viewpoint. Paired sampling is a technique that is used very often in practice, so this part is especially meaningful.

- **Methodological/Experimental strengths**

Since the method uses the Fourier basis and focuses the computational budget on odd-order interactions, it shows very strong performance. The empirical message is also solid. The paper does not only show that OddSHAP performs well, but also provides an ablation study on the choice of the number of (odd-order) interactions, which is helpful for users who may want to choose the number of interaction terms in practice.

## Weaknesses

- **Missing Crucial Prior work and Benchmark Comparison**

Most importantly, the fact that even-order interactions do not affect the determination of the Shapley value in the Fourier basis, and therefore the budget should be focused on odd-order interactions, is not entirely new knowledge. Zhou et al. (2025) already studied this mathematically in the statistics literature. Based on the classical statistics area of fractional factorial designs, they showed in the language of their field that the utility function can be written in the span of the Fourier basis, and that even-order interactions become zero when computing the Shapley value.

Specifically, they assume the setting where each player’s (non-)participation in the game is represented by a factor in $\\{-1,1\\}$, and the utility function is defined as the expectation of a two-level factorial experiment. That is, if $\boldsymbol{x}_S = (x_1(S),\dots,x_n(S))$ where $x_i(S)=2\mathbb{I}\\{i\in S\\}-1$, then

$$y(\\boldsymbol{x}\_S) = \\sum\_{T\\subseteq [n]}\\left(\\prod_{i\\in T}x_i(S)\\right)\\beta_T+\\epsilon$$

is defined (Equation (3) in their paper), and they set $\mathbb{E}(y(\boldsymbol{x}_S))=U(S)$ (Equation (4) in their paper). Then, through their Theorem 1, they show that the case where $|T|$ is even does not affect the Shapley value. Based on this observation, Zhou et al. (2025) make a ``truncation assumption”, namely that $\beta_T = 0$ when $|T|\ge 5$. In other words, they assume that interactions above a certain order do not affect the determination of the Shapley value. Then, if all budget is focused on the cases $|T|=1$ and $|T|=3$, they show that the ground truth can be computed with only $4n^2-4$ utility evaluations when the truncation assumption holds, and that even when it does not hold, a low approximation error can still be achieved by post-hoc correction.

In that sense, this submission and Zhou et al. (2025) seem to be based on almost the same philosophy, even though they use different language. In particular, the statement that even-order interactions do not matter is mathematically equivalent as well. This is because the FFD formula itself can be viewed as an expansion in the Fourier basis. In particular, if we reverse the factorization used by Zhou et al. in FFD and define $\\tilde{x}\_i(S):=-x\_i(S)=1-2\\mathbb{I}\\{i\\in S\\} = (-1)\^{\\mathbb{I}\\{i\\in S\\}}$, then $\prod_{i\in T} \tilde{x}_i(S)=(-1)^{|S\cap T|}=\chi_T(S)$. So, after only adjusting the sign of the $\beta$’s, the FFD regression model can be understood as expanding the utility function in the Fourier basis. Furthermore, considering this reparametrization, Theorem 1 in Zhou et al. (2025) itself is equivalent to the formula that recovers the Shapley value from the regression representation with higher-order terms through Harsanyi dividends, that is, Equation (5) and hence (14) in this submission. In terms of methodological strategy, both works are motivated by the idea of spending their computational budget only to where it determines Shapley value.

At the same time, I also believe that the novelty of this submission is not limited only to that point, because it also gives a justification for paired sampling and connects that idea to a practically strong estimator. Still, Observation 3.1 and Corollary 3.3, which is presented in this submission as “new knowledge”, has already been studied before, and this should affect the assessment of originality.

In terms of presentation, I acknowledge that the authors may have reached the same conclusion independently without knowing the earlier work. Even so, considering the submission and publication timeline of this paper and Zhou et al. (2025), I do not think it is reasonable to present the authors as co-discoverers of this idea. For that reason, I believe the authors should position the novelty of their paper more carefully. Also, Zhou et al. (2025) still seems to provide a valid estimator that can be directly used for SHAP problems in machine learning. So, I think this FFD-Shapley estimator should also be included in the experiments as a strong benchmark.

Again, I still recognize the novelty of this work even aside from this observation. However, in its current form, I do not think the paper is acceptance-ready because of the missing reference to, and comparison with, such a closely related work. If the authors clarify their positioning on this point and compare against this existing method through additional experiments, I would be willing to raise my rating.

- **Mixed Theoretical Levels**

As one more minor point, the presentation around paired sampling and odd-even decomposition is actually not very easy to understand. These ideas are somewhat mixed together. As I understand it, the advantage of paired sampling comes from the fact that odd-even decomposition is always possible. On the other hand, in the Fourier basis, the basis set itself can actually be partitioned into odd and even parts, while this is not possible in the unanimity basis set. Therefore, the theoretical justification of paired sampling itself seems to hold regardless of the basis choice, but OddSHAP itself, that is, the actual method that focuses on odd interactions, requires the Fourier basis. I think these two levels should be explained more clearly and separately.


**Reference**

- Zhou, Z., Mee, R., Hamers, H., & Zheng, W. (2025). Fast Approximation of Shapley Values through Fractional Factorial Designs. *Journal of the American Statistical Association*, 1-20.

---

> ### Author Rebuttal · Authors · 2026-03-30
>
> We thank the reviewer for their thorough and constructive feedback. We address the concerns below, and include additional experiments:
> - Added baselines: https://shorturl.at/lDAYy
> - Added ablations: https://shorturl.at/lOkTy
>
> > Missing Crucial Prior work and Benchmark Comparison:
>
> [1] Zhou et al. (2025). Fast Approximation of Shapley Values through Fractional Factorial Designs
>
> We were not aware of [1]. We agree that it follows the same philosophy and will update our submission to properly highlight our unique contributions. Specifically,
> - Observation 3.1 will not be a claimed contribution. We will instead cite [1] and mention how the result for $f_{odd}$ follows.
> - Our Lemma 3.4 follows by Theorem 5 of [1]. We will remove it and cite accordingly.
> - We still believe Corollary 3.3 and Theorem 3.5 are novel contributions, focusing on the implications of paired sampling under the regression objective.
>
> In Figure 1 (link 1), we added the methods of [1] to our approximation quality experiment. Dark purple is their Algorithm 1, and light purple is their Algorithm 2 (bias-corrected). For ViT and DistilBERT, FFD-RD is outperformed by OddSHAP. For tree-based value functions FFD-RD outperforms OddSHAP on Estate, Cancer, IL60, and CG60, while having comparable performance on NHANES and Crime. Empirically, we find their bias correction step generally degrades performance (except ViT).
>
> In Figure 2 (link 1), we explored the truncation assumption of [1] by computing the proportion of spectral energy (sum of squared Fourier coefficients) by degree for each task (using the Fourier sparse recovery algorithm SPEX under large sample budget). 5+-order energy is practically absent in tree-based value functions but present in ViT and DistilBERT. This finding explains the strong performance of FFD-RD on the tree-based value functions and the poor performance on DistilBERT and ViT. We will include the algorithms of [1] as strong baselines in our experiments and add a discussion.
>
> > How do you posit the novelty of the observation that even-order interactions do not matter, especially in relation to [1]?
>
> This observation is not novel given [1]. We will reposition as such.
>
> > Compared with the FFD-based estimator of [1], what are the exact theoretical or practical advantages of OddSHAP?
>
> We see two advantages of OddSHAP over the FFD-RD estimator:
> 1. **OddSHAP uses flexible budgets**, while FFD-RD only works for a single budget with $\mathcal O(d^2)$ samples. Given real world resource constraints, estimators with *variable* budgets are important for practitioners computing Shapley value.
> 2. **OddSHAP is able to capture odd-interactions beyond order 3**, which our empirical result suggests is critical for achieving SOTA estimation quality in value functions with higher-order interactions (DistilBERT,ViT). Our results suggest the Bias correction algorithm of [1] is insufficient.
>
> > Could you explain more clearly which parts require the Fourier basis and which parts do not?
>
> Theorem 3.2 holds independent of the basis because it considers a function class $\mathcal F$. However, to exploit Theorem 3.2, we need to fit odd functions, which will then be exclusively fit on $f_{odd}$ through paired sampling. In OddSHAP, to avoid the time complexity of fitting the irrelevant even functions, we achieve this by selecting odd-cardinality basis vectors from the Fourier basis. This is **specific** to the basis and a strong benefit over PolySHAP relying on the unanimity basis.
>
>  > How sensitive is the method to this term selection and to hyperparameters such as tree depth or the number of trees? Could you provide a small sensitivity study?
>
> In Figure 3 and 4 (link 2), we compared the MSE ratio for max-depth and n-trees with OddSHAP using 10/100. Restricting max-depth and n-trees has a negative impact, while increasing n-trees rarely improves approximation quality. We observe that the default parameters work well.
>
> > Could you present the full runtime more clearly, including GBT training, interaction extraction, regression solve, and SHAP computation? In particular, a runtime comparison with PolySHAP and RegressionMSR at the same $m$ and $d$ would be useful.
>
> In Figure 6 (appendix), we show the full runtime by each step of OddSHAP. Figure 5 shows the runtime for different $m$ across all datasets and methods excluding evaluation costs. Figures 7-10 showcase the runtime under different evaluation costs.
>
> > While reviewing this paper among the papers assigned to me, I found an overlap with another submission I was assigned [...]
>
> One of the authors on our work is also an author on the other submission. While writing the related work, they copied and reworded the paragraph in question. Even though the author is on both works, we understand this is not okay and we will rewrite the related works.

---

> > ### Author Rebuttal · Reviewer_u3wk · 2026-04-02
> >
> > Thank you very much for the really sincere and thoughtful rebuttal. I clearly understand that I mainly raised two concerns, 1) existing benchmarks which already address what this submission addresses, and 2) potential rewriting of another submission, and that these concerns may have been burdensome for the authors. Even so, authors fully addressed my concerns. In particular, for 1), authors honestly repositioned the novelty of the paper, and for 2), they also answered the rewriting issue in an honest way. In fact, I already thought this paper is strong and solid, and my previous harsh rating was only to correct these issues.
> >
> > At this point, compared with the FFD methodology, I acknowledge that the flexibility in budget setting and the ability to include higher-order terms are indeed practical strengths. Also, regarding the rewriting issue, I may have been quite sensitive, but I raised this issue because I believed that if something is not right, then it should be properly corrected, and I truly appreciate that you addressed it well.
> >
> > I feel that the new knowledge presented by this paper is impactful even considering the existence of prior related work, so I am raising my score and support acceptance of this paper.

---

### Official Review · Reviewer_BhiB · 2026-03-11

**Soundness:** 4
**Presentation:** 3
**Significance:** 2
**Originality:** 4
**Overall Recommendation:** 4
**Confidence:** 1

**Summary:**

The authors consider the problem of estimating Shapley values, given only query access to the utility function.

With Shapley values, increasingly being used for credit assignment for both features and data, the question of estimating Shapley values is central to modern machine learning.
Many data Shapley applications utilize white-box heuristics such as using influence functions, but researching better black-box methods still has many applications from feature Shapley, to validating the heuristic methods, to providing an alternative to the white-box methods.

The authors propose OddSHAP -- a Shapley value estimator which utilizes their observation that only odd Fourier coefficients of the utility function can affect the Shapley to improve upon previous Shapley estimations which were based on using a linear regression to fit a low-degree approximation to the utility function.

The authors prove that OddSHAP converges to the true Shapley value as the compute budget goes to infinity and show experimentally that it converges faster on some real-world datasets in some compute budgets.

The observation behind OddSHAP also further motivates the previously observed phenomena that paired sampling yields better approximation for the Shapley values.

**Compliance With Llm Reviewing Policy:**

Affirmed.

**Key Questions For Authors:**

See questions above.

**Limitations:**

yes

**Strengths And Weaknesses:**

The paper is well-written, and seems sound to me (though it is not entirely in my domain).

As a reader with limited background in this specific field, it is hard for me to get a sense of the significance of OddSHAP, primarily because I did not see an analysis of its convergence compared to standard methods like FourierSHAP or MSR (when the main contribution seems to be that OddSHAP converges with fewer samples).

At a very high-level, it seems that OddSHAP differs from these methods in that we can skip even-degree contributions, so intuitively, I would expect methods like FourierSHAP to work well for degree $\leq k$ utility functions if they have a compute budget of that scales with $\binom{d}{\leq k} \approx \binom{d}{k}$, and if we are then also told that even-degree contributions do not affect the Shapley value, this allows us to limit ourselves to the odd-degree terms in $\binom{d}{\leq k}$.

If we are further given that $k$ is even and much smaller than $d$, that would allow us to skip all the degree $=k$ terms, which should give us a runtime that now scales with $\binom{d}{<k} \approx \binom{d}{k-1}$.
This intuition seems to align with the observation that OddSHAP only beats existing methods when $d$ is sufficiently large and seems to alternate with a different technique for moderate $d$ (Figure 2).

1. Does this intuition make sense / is there some other intuition for why and when OddSHAP would qualitatively outperform existing techniques?

2. Is there a (potentially very restricted) setting where you can prove that OddSHAP converges faster than existing techniques?

3. Finally, I didn't follow the role of the ODDINTERACTIONEXTRACT subroutine in Algorithm 1. Could the authors elaborate on this step?

---

> ### Author Rebuttal · Authors · 2026-03-30
>
> We thank the reviewer for their thoughtful questions and appreciation for our work. We address the questions and concerns below, including additional experiments:
> - Added baselines: https://shorturl.at/lDAYy
> - Added ablations: https://shorturl.at/lOkTy
>
> > Does this intuition make sense / is there some other intuition for why and when OddSHAP would qualitatively outperform existing techniques?
>
> At a very high level, we can group existing approximation methods into two groups:
> Group 1: Use Monte Carlo sampling to directly estimate the Shapley value, no interactions are estimated and only implicitly considered (MSR, SVARM, and PermutationSHAP).
> Group 2. Fit a proxy $\hat f$ to the value function $f$, such that the Shapley values of the proxy can be efficiently extracted/approximated.
>
> Within Group 2, we can further distinguish:
> - FourierSHAP assumes that $f$ has a sparse fourier representation and $\hat f$ can be found through recovering this sparse support of influential interactions.
> - Tree-based proxies (Proxy, RegressionMSR) fit a tree-based proxy, and extract the Shapley values via TreeSHAP.
> - KernelSHAP and LeverageSHAP use a linear proxy (without interactions), where the Shapley values of the proxy are simply the coefficients of the linear model. Here, paired sampling has shown substantial performance benefits.
> - PolySHAP uses a polynomial proxy with **random/all** interactions in the unanimity basis for which the Shapley value can be efficiently extracted by distributing each coefficient among the involved variables.
>
> Our OddSHAP uses a polynomial proxy in the **Fourier basis** using influential odd Fourier interactions. It can be viewed as
>   - an extension of sparse Fourier proxies because we too assume that $f$ is well-approximated with a sparse Fourier support.
>   - an extension of PolySHAP that selects interaction terms for the polynomial proxy using a **better basis**.
>   - an extension of tree-based approaches, since we select these interactions based on the tree-based proxy using ProxySPEX.
>
> The benefit over FourierSHAP is that their sparse recovery algorithm focuses on learning a proxy $\hat f$ that minimizes $\ell_2$ loss (i.e. recovering the highest magnitude Fourier coefficients of $f$). For Shapley estimation, this loss objective is misaligned (e.g. even Fourier coefficients do not affect Shapley estimation). Our sampling strategy, support of only odd interaction, and regression accounts for this.
>
> The benefit over PolySHAP is that we can exclude all even-order interaction terms and focus on the most influential Fourier interactions, overcoming the combinatorial explosion of interaction terms.
>
> The benefit over tree-based proxies (except for RegressionMSR) is that we still obtain a **consistent** estimator by using the polynomial regression approach while retaining the benefits of the implicit regularization of trees to efficiently select important interactions. OddSHAP uses only **only odd** interactions, while tree-based proxies implicitly fit even and odd interactions.
>
> Overall, this unique combination shows substantial benefits in approximations. In Figure 2 (link 1), we added the FourierSHAP baseline, which performs among the worst baselines. The main competitors and SOTA approaches for Shapley value estimation are currently LeverageSHAP and RegressionMSR.
>
> >  Is there a (potentially very restricted) setting where you can prove that OddSHAP converges faster than existing techniques?
>
> Consider a value function $f$ with exactly $\ell$ odd-cardinality Fourier interactions and arbitrary even-cardinality Fourier interactions. Moreover, assume that ProxySPEX is able to identify these top-$\ell$ odd-cardinality interaction terms. In this case OddSHAP with $\ell$ interaction terms is guaranteed to exactly recover the ground-truth Shapley values of $f$ with $m>d+\ell$ samples.
>
> KernelSHAP and PolySHAP would recover the exact Shapley values only with $2^d$ samples. PolySHAP could recover the exact Shapley values also with $m>d+\ell’$ samples, where $\ell’ \approx 2^{\vert S_{max}\vert}$ is the number of interaction in the unanimity basis and $S_{max}$ the Fourier interaction with the highest cardinality. All tree-based proxies recover the exact Shapley values only if all $\ell$ odd interactions are exactly fit within the tree. Note that this task is substantially more difficult than identifying the top-$\ell$ interactions. We will add this discussion in more detail to the appendix.
>
> > Finally, I didn't follow the role of the \textsc{OddInteractionExtract} subroutine in Algorithm 1. Could the authors elaborate on this step?
>
> The whole point of this step is that we need some way of identifying the *important* odd interactions in the Fourier basis. Fortunately, there have been several recent works that do just this: We use ProxySPEX (Butler et al., 2025), which fits a GBT and extracts the largest absolute value interactions from the trees. We will add a more detailed explanation in the appendix.

---

> > ### Author Rebuttal · Reviewer_BhiB · 2026-04-02
> >
> > The authors have addressed all of my concerns.
> >
> > The paper seems like a good ICML contribution to me (estimating Shapley values even for small models given black-box access to their utilities is very hard in practice, and any method that gives a significant and explainable practical improvement over the SOTA by combining both a new theory and algorithm with proper empirical validation seems like a very significant contribution to me), and I plan to increase my rating and recommend accepting it.
> >
> > However, since the exact problem is outside of my domain, I do not feel qualified to judge the issues raised by other reviewers regarding comparison with prior works, and leave that to the AC / the discussions with other reviewers.

---

### Decision · Program_Chairs · 2026-04-30

**Decision:**

Accept (regular)

**Comment:**

All reviewers appreciate the theoretical analysis of paired sampling in Shapley-value computation. While widely in practice, it thusffar lacked formal underpinnings for why it works. The authors here elucidate why paired sampling can have a positive effect (wrt approximation quality) which is a nice contribution. At the same time, the practical results and rebuttal show there also exists a negative effect, which is not discussed in sufficient detail let alone characterized; therewith it is also not clear when the positive effect outweighs the negative one, or whether we should simply hope for the best. The rebuttal was a positive exchange that helped to clarify a number of concerns. Overall, I'm happy to recommend acceptance, assuming the authors fix the error in the proof of Corr. 3.3, and include a discussion on the negative effects of paired sampling.